# $f$-Mutual Information Contrastive Learning

## Abstract

Self-supervised contrastive learning is an emerging field due to its power in providing good data representations. Such learning paradigm widely adopts the InfoNCE loss, which is closely connected with maximizing the mutual information. In this work, we propose the $f$-Mutual Information Contrastive Learning framework ($f$-MICL), which directly maximizes the $f$-divergence-based generalization of mutual information. We theoretically prove that, with mild assumptions, our $f$-MICL naturally attains the *alignment* for positive pairs and the *uniformity* for data representations, the two main factors for the success of contrastive learning. We further provide theoretical guidance on designing the similarity function and choosing the effective $f$-divergences for $f$-MICL. Using several benchmark tasks from both vision and natural language, we empirically verify that our novel method outperforms or performs on par with state-of-the-art strategies.

## 1 Introduction

Contrastive learning has attracted a surge of attention recently due to its success in learning informative representation for image recognition, natural language understanding, and reinforcement learning (Chen et al., 2020; He et al., 2020; Logeswaran & Lee, 2018; Srinivas et al., 2020). Such learning paradigm is fully unsupervised by encouraging the contrastiveness between similar and dissimilar sample pairs. Specifically, the feature embeddings of similar sample pairs are expected to be close while those of dissimilar sample pairs are expected to be far apart. To attain this goal, a softmax cross-entropy loss, a.k.a. InfoNCE, has been widely used (Wu et al., 2018; van den Oord et al., 2018; Chen et al., 2020; Hénaff et al., 2020; He et al., 2020), which aims to maximize the probability of picking a similar sample pair among a batch of sample pairs.

InfoNCE can be interpreted as a lower bound of the mutual information (MI) between two views of data samples (van den Oord et al., 2018; Bachman et al., 2019; Tian et al., 2020a; Tschannen et al., 2020). This explanation is consistent with the well-known "InfoMax principle" (Linsker, 1988). Nevertheless, it has been shown that maximizing a tighter bound on the MI can result in worse representations (Tschannen et al., 2020); and reducing the MI between views while only keeping task-relevant information can improve the downstream performance (Tian et al., 2020b). These observations suggest that maximizing the MI may be insufficient in contrastive learning and thus a better objective design is required.

To attain the aforementioned goal, we propose a novel contrastive learning framework, coined as $f$-MICL. In a nutshell, leveraging the fact that MI can be formulated as the Kullback–Leibler (KL) divergence between the joint distribution and the product of the marginal distributions, we replace the KL divergence with the general $f$-divergence family (Ali & Silvey, 1966; Csiszár, 1967). Doing so, we obtain a generalization of mutual information, called *$f$-mutual information* ($f$-MI, Csiszár, 1967). Notably, by maximizing a lower bound of $f$-mutual information we naturally decompose the objective function into two terms, which correspond to the properties of the *alignment* and the *uniformity*. Such characterization has been revealed in Wang & Isola (2020, Theorem 1) for the InfoNCE loss. Compared with Wang & Isola (2020), our result applies to a wide range of the $f$-divergence family such as KL, Jensen–Shannon (JS), Pearson $\chi^2$ and Vincze–Le Cam. and it does not rely on the limit of an infinite number of dissimilar samples. This allows us to explore the space of $f$-MI and improve the performance of InfoNCE-based contrastive learning.

The similarity function is crucial for the evaluation of the contrastiveness of similar and dissimilar sample pairs. Commonly used similarity functions include the cosine similarity (Chen et al., 2020;

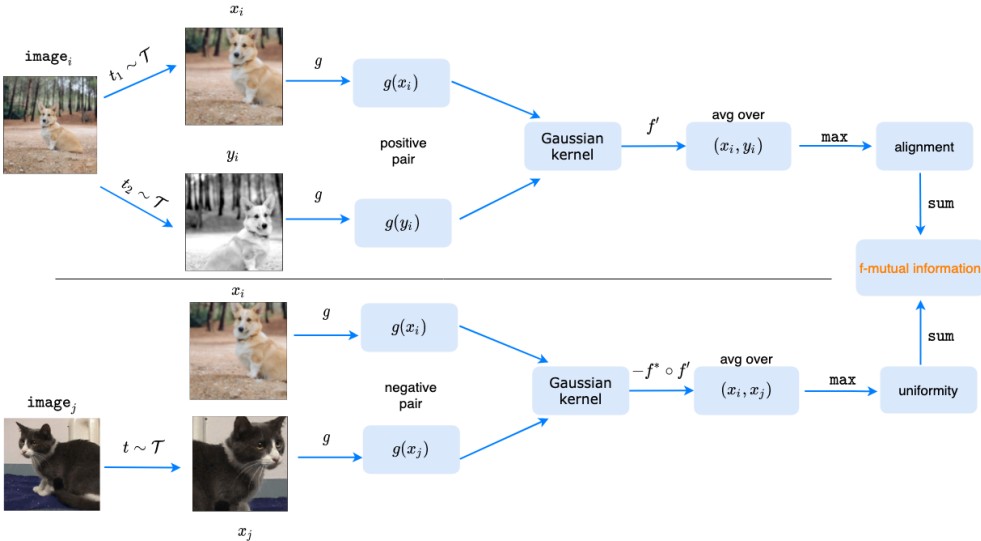

Figure 1: Network architecture of our proposed $f$-MICL. $\texttt{image}_i$: the $i^{\text{th}}$ image in the current batch; $f$: the function used in the $f$-mutual information (§2); $g$: feature embedding; $t$, $t_1$, $t_2$: augmentation functions drawn from the same family $\mathcal{T}$ of augmentations; $f'$: the derivative; $f^*$: the Fenchel conjugate. The symbol $\circ$ denotes function composition. The sum of the two terms gives the variational lower bound of $f$-mutual information. See equation 10 for more details.

He et al., 2020), the bilinear functions (van den Oord et al., 2018; Tian et al., 2020a; Hénaff et al., 2020), and the neural network based scores (Hjelm et al., 2018). While most aforementioned similarity functions for contrastive learning are heuristic and pre-designed, in this work, we provide a principled way to design the similarity function. By assuming that the joint feature distribution of two similar samples is proportional to a Gaussian kernel, we derive an optimal similarity function for practical use, which resembles the well-known radial basis functions (Powell, 1987).

Figure 1 gives a high-level summary of our $f$-MICL framework. Given a batch of samples (*e.g.*, images) we generate *positive pairs* via data augmentation and *negative pairs* using other augmented samples in the current batch. With the optimization of our $f$-mutual information objective the positive pairs are aligned with each other and the data representations are uniformly distributed. Our contributions can be summarized as follows:

- We propose a novel framework for contrastive learning (called $f$-MICL) by encouraging the contrastiveness of positive and negative pairs with a general $f$-divergence family.
- With an assumption on the joint feature distribution we provide an optimal design for the similarity function with Gaussian kernels, which shows interesting connection between contrastive learning and kernel methods.
- We characterize the properties of alignment and uniformity for $f$-MICL and provide guidance on choosing proper $f$-divergences. Our $f$-MICL objective can be estimated with finite samples and we give the corresponding error bound.
- Experimentally, our framework is better or on par with popular baselines, by simply replacing, *e.g.*, the InfoNCE loss function with our theoretically grounded $f$-MICL objectives.

**Notations** We assume a dominating measure $\lambda$ (e.g. Lebesgue) is given and all other probability measures are represented as some density w.r.t. $\lambda$. We denote $D_f(p\|q)$ as the $f$-divergence between two densities functions $p$ and $q$. Given the joint density $p(x, y)$, we denote $p(x) = \int p(x, y)\mathrm{d}\lambda(y)$ and $p(y) = \int p(x, y)\mathrm{d}\lambda(x)$ as the marginals. We use $\mathrm{supp}(\cdot)$ to denote the support of a distribution, and $f^*$ to denote the Fenchel conjugate of a convex function $f$. Every norm presented is Euclidean.

## 2 Preliminaries

We provide some preliminaries for our framework in the context of contrastive learning. Contrastive learning is a popular *unsupervised* method for learning data representations. In contrastive learning,

Table 1: A summary of common $f$-divergences. KL: Kullback–Leibler; JS: Jensen–Shannon; and SH: Squared Hellinger. For JS, we define $\varphi(u) = -(u+1)\log\frac{1+u}{2} + u\log u$. For the Pearson $\chi^2$, we take $f^*(t) = -1$ if $t \leq -2$. The Tsallis-$\alpha$ divergence is defined in Tsallis (1988) and we have $\alpha > 1$ for $f$-divergences. The Vincze–Le Cam (VLC) divergence can be found in Le Cam (2012, p.47), which is closely related to the Pearson $\chi^2$ and Hellinger divergences. For the Vincze–Le Cam divergence we require $-3 < t < 1$ and $f^*(t) = -1$ if $t \leq -3$.

| Divergence | $f(u)$ | $f^*(t)$ | $f'(u)$ | $f^* \circ f'(u)$ |
|---|---|---|---|---|
| KL | $u\log u$ | $\exp(t-1)$ | $\log u + 1$ | $u$ |
| JS | $\varphi(u)$ | $-\log(2-e^t)$ | $\log 2 + \log\frac{u}{1+u}$ | $-\log\frac{2}{1+u}$ |
| Pearson $\chi^2$ | $(u-1)^2$ | $t^2/4 + t$ | $2(u-1)$ | $u^2 - 1$ |
| SH | $(\sqrt{u}-1)^2$ | $\frac{t}{1-t}$ | $1 - u^{-1/2}$ | $u^{1/2} - 1$ |
| Tsallis-$\alpha$ | $u^\alpha/(\alpha-1)$ | $((\alpha-1)t/\alpha)^{\alpha/(\alpha-1)}$ | $\frac{\alpha u^{\alpha-1}}{\alpha-1}$ | $u^\alpha$ |
| Vincze–Le Cam | $\frac{(u-1)^2}{u+1}$ | $4 - t - 4\sqrt{1-t}$ | $1 - \frac{4}{(u+1)^2}$ | $3 - \frac{4}{u+1}$ |

we expect that similar sample pairs to be close to each other in the embedding space while uncorrelated pairs to be far away. Denote $p_{\text{pos}}$ as the distribution of *positive pairs*, *i.e.*, two samples that have similar representations. Assume that this distribution is symmetric w.r.t. the two random variables, then the resultant two marginals both follow the data distribution $p_{\text{data}}$ (Wang & Isola, 2020).

In this work we propose the $f$-mutual information framework for contrastive learning. First recall the $f$-mutual information ($f$-MI) between a pair of random variables $X$ and $Y$:

**Definition 1** ($f$**-mutual information, Csiszár 1967**)**.** *Consider a pair of random variables $(X, Y)$ with density function $p(x, y)$. The $f$-mutual information $I_f$ between $X$ and $Y$ is defined as*

$$I_f(X;Y) := D_f\left(p(x,y)\|p(x)p(y)\right) = \int f\left(\frac{p(x,y)}{p(x)p(y)}\right) p(x)p(y) \cdot \mathrm{d}\lambda(x,y), \qquad (1)$$

*where $f : \mathbb{R}_+ \to \mathbb{R}$ is (closed) convex with $f(1) = 0$, and recall that $p(x)$ and $p(y)$ are the marginal densities of $p(x, y)$, whereas $\lambda$ is a dominating measure (e.g. Lebesgue).*

Common choices of $f$ can be found in Table 1 and Table 4 (Appendix A), which we will discuss in more details in §3. It is well-known that $f$-mutual information is non-negative and symmetric, and provided that $f$ is strictly convex, $I_f(X;Y) = 0$ iff $X$ and $Y$ are independent (Ali & Silvey, 1966). When $X$ and $Y$ are of high dimension, it is quite challenging to estimate the $f$-divergence directly. Instead, Nguyen et al. (2010) derived a variational method by maximizing the dual problem:

$$I_f(X;Y) \geq \sup_{T\in\mathcal{T}} i_f(X;Y) := \mathbb{E}_{(x,y)\sim p_{\text{pos}}}[T(x,y)] - \mathbb{E}_{(x,y)\sim p_{\text{data}}\otimes p_{\text{data}}}[f^*(T(x,y))], \qquad (2)$$

where $f^*(t) := \sup_{x\in\mathbb{R}_+}(xt - f(x))$ is the (monotone) Fenchel conjugate of $f$, and is always *monotonically increasing*. Here $\mathcal{T}$ is a class of functions $T : \text{supp}(p_{\text{data}}) \times \text{supp}(p_{\text{data}}) \to \text{dom } f^*$. Nguyen et al. (2010) showed that the bound in eq. (2) is tight if there exists $T^* \in \mathcal{T}$ such that

$$T^*(x,y) = f'\left(p(x,y)/(p(x)p(y))\right), \text{ for any } (x,y) \in \text{supp}(p_{\text{data}}) \times \text{supp}(p_{\text{data}}), \qquad (3)$$

and in particular, if $\mathcal{T}$ comprises of all (measurable) functions.

## 3 DESIGN OF $f$-MICL

Based on the $f$-MI introduced in §2, we propose a novel framework for contrastive learning. Furthermore, we characterize the properties of alignment and uniformity theoretically for general $f$-divergences. Following Chen et al. (2020), we design the strcture of function $T$ as follows:

$$T(x,y) := k(g(x), g(y)), \text{ where } \|g(x)\| = 1 \text{ for any sample } x. \qquad (4)$$

The function $g$ produces a $d$-dimensional normalized feature encoding on the hypersphere $\mathbb{S}^{d-1}$ and $k$ is a similarity function that measures the similarity between two embeddings $g(x)$ and $g(y)$. With the above interpretation, we can rewrite our objective of $f$-mutual information, equation 2, as:

$$\sup_{g\in\mathcal{G}, k\in\mathcal{K}} i_f(X;Y) := \mathbb{E}_{(x,y)\sim p_{\text{pos}}}[k(g(x),g(y))] - \mathbb{E}_{(x,y)\sim p_{\text{data}}\otimes p_{\text{data}}}[f^*(k(g(x),g(y)))], \qquad (5)$$

where $\mathcal{G}$ and $\mathcal{K}$ are the function classes of the feature encoder $g$ and the similarity function $k$. We can treat the first term as the similarity score between *positive pairs* in the feature space, and the second term as the similarity score between two random samples, a.k.a. *negative pairs*, in the feature space. As $f^*$ is increasing, maximizing $f$-MI is equivalent to simultaneously maximizing the similarity between positive pairs and minimizing the similarity between negative pairs.

### 3.1 OPTIMIZED SIMILARITY FUNCTION AND IMPLEMENTATION

Let us now study how to search for the optimal similarity function $k$. To our best knowledge, there has been no theoretical study on the choice of similarity functions. Most existing contrastive learning methods (e.g. Chen et al., 2020; Tsai et al., 2020; He et al., 2020) adopt a pre-designed similarity function, such as the cosine similarity. For the ease of notation, from now on we define $x^g := g(x)$ and $y^g := g(y)$. Suppose $(x, y) \sim p_{\text{pos}}$, then we denote $p_{\text{pos}}^g$ as the distribution of $(x^g, y^g)$, and $p_{\text{data}}^g$ as the marginal feature distribution of $x^g$ or $y^g$. The corresponding density functions are written as $p_g(x^g), p_g(y^g)$ and $p_g(x^g, y^g)$. We remind the reader of the following result:

**Lemma 2** (*e.g.*, Nguyen et al. 2010, Lemma 1). *Suppose $f$ is differentiable, and the encoder function $g$ is fixed. The similarity function*

$$k^*(x^g, y^g) = f'\left( \frac{p_g(x^g, y^g)}{p_g(x^g)p_g(y^g)} \right) \tag{6}$$

*maximizes $i_f(X; Y)$ in eq. (5) as long as it is contained in the function class $\mathcal{K}$.*

Equation 6 provides an optimal similarity function, which nevertheless depends on the density functions. Comparing equation 6 with equation 5, we realize that the optimal $k^*$ in fact gives the $f$-MI on the feature space, $I_f(g(X), g(Y))$, which is a low bound of the original $f$-MI, $I_f(X; Y)$. To use $k^*$ practically we make the following assumption on the joint density:

**Assumption 3.** *The joint feature distribution is proportional to a radial basis function (RBF), i.e.,*

$$p_g(x^g, y^g) \propto \varphi(\|x^g - y^g\|^2) \text{ for a real-valued function } \varphi.$$

Radial basis functions are widely used in kernel methods (Powell, 1987; Murphy, 2012), and the Gaussian kernel is perhaps the most well-known RBF. Throughout this work we mainly consider $\varphi$ as a Gaussian kernel:

$$\varphi(\|x^g - y^g\|^2) = G_\sigma(\|x^g - y^g\|^2) := \mu \exp\left( -\frac{\|x^g - y^g\|^2}{2\sigma^2} \right), \tag{7}$$

with $\mu$ a constant left to be determined. Fixing $y^g$, then $p_g(\cdot, y^g)$ is known as the *von Mises–Fisher* distribution (von Mises, 1918; Fisher, 1953; Bingham & Mardia, 1975), since $x^g$ and $y^g$ are unit vectors. With Assumption 3 on the joint density, the resultant marginal feature distribution $p_{\text{data}}^g$ is uniform on the hypersphere $\mathbb{S}^{d-1}$, where $d$ is the dimension of the feature space (see Prop. 8 in App. B). Additionally, for positive pairs the distance in the feature space, $\|x^g - y^g\|$, is more likely to be small. If the variance $\sigma^2 \to 0$, then the Gaussian kernel becomes the Dirac delta distribution, $\delta_{x^g = y^g}$. This requires that the two features $x^g$ and $y^g$ to be the same, which is desirable. In general, the radial basis function $\varphi$ should be decreasing since a positive pair should be more likely to be adjacent in the feature space. For example, with $\varphi(t) = 1 - t/2$ we obtain the cosine similarity.

Based on Assumption 3 we propose the following similarity function between pairs of features:

**Theorem 4** (**Gaussian similarity**). *Under Assumption 3 with Gaussian kernels and the same settings as Lemma 2, the optimal similarity function $k^*$ satisfies that for any $x^g, y^g \in \mathbb{S}^{d-1}$:*

$$k^*(x^g, y^g) = f'(CG_\sigma(\|x^g - y^g\|^2)), \tag{8}$$

*where $d$ is the feature dimension and $C$ is an absolute constant.*

For simplicity we will rewrite $k^*(x^g, y^g) = f' \circ G_\sigma(\|x^g - y^g\|^2)$ by absorbing the constant $C$ into $G_\sigma$, since we have left some flexibility in equation 7. Although Assumption 3 with Gaussian kernels may not always reflect the real feature distribution, we can still use the similarity function in equation 8, even if it might not be optimal. In our experiments in §5, the Gaussian similarity equation 8 consistently outperforms the default cosine similarity in contrastive learning.

---

**Algorithm 1:** $f$-mutual information contrastive learning ($f$-MICL)

---

**Input:** batch size $N$, function $f$, weighting parameter $\alpha$, constant $\mu$ (in $G_\sigma$), variance $\sigma^2$, optimizer

1 **for** *each batch* $\{z_i\}_{i=1}^N$ **do**

2      **forall** $k \in [1, N]$ **do**

3          randomly sample two augmentation functions $t_1, t_2$

4          $y_k \leftarrow t_1(z_k)$, $x_k \leftarrow t_2(z_k)$

5      compute $i_f = \frac{1}{N} \sum_{i=1}^N \left[ f' \circ G_\sigma(\|x_i^g - y_i^g\|^2) \right] - \frac{\alpha}{N(N-1)} \sum_{i \neq j} f^* \circ f' \circ G_\sigma(\|x_i^g - x_j^g\|)$

6      update $g$ by taking a step to maximizing $i_f$ using the optimizer

---

Bringing the optimal $k^*$ in equation 8 into our objective equation 5 we have the following objective:

$$\sup_{g \in \mathcal{G}} \mathbb{E}_{(x,y) \sim p_{\text{pos}}} \left[ f' \circ G_\sigma(\|x^g - y^g\|^2) \right] - \mathbb{E}_{(x,y) \sim p_{\text{data}} \otimes p_{\text{data}}} \left[ f^* \circ f' \circ G_\sigma(\|x^g - y^g\|^2) \right], \quad (9)$$

where $G_\sigma$ is defined in equation 7. With a similar sampling method of positive and negative pairs in Chen et al. (2020), given a batch of $N$ samples we can estimate the objective in equation 9 as:

$$\widehat{i}_f(X; Y) = \frac{1}{N} \sum_{i=1}^N f' \circ G_\sigma(\|x_i^g - y_i^g\|^2) - \frac{1}{N(N-1)} \sum_{i \neq j} f^* \circ f' \circ G_\sigma(\|x_i^g - x_j^g\|^2), \quad (10)$$

where $x_i$ and $y_i$ are two different kinds of data augmentation of the $i$-th sample, and $x_i$ and $x_j$ are different samples of the same kind of data augmentation.

With the objective in equation 10 we propose our algorithm for contrastive learning in Algorithm 1. Note that we treat $\mu$ and $\sigma^2$ in our Gaussian kernel equation 7 as hyperparameters. To balance the two terms in our objective, we additionally include a weighting parameter $\alpha$ in front of the second term. We can prove that rescaling the second term with the factor $\alpha$ is equivalent to changing the function $f$ to another convex function $f_\alpha$ (see Prop. 7 in Appendix A).

### 3.2 ALIGNMENT AND UNIFORMITY

Notably, if we choose the $f$-divergence to be the KL divergence, the objective in equation 9 becomes:

$$\sup_{g \in \mathcal{G}} -\frac{1}{2\sigma^2} \mathbb{E}_{(x,y) \sim p_{\text{pos}}} \left[ \|x^g - y^g\|^2 \right] - \mu \mathbb{E}_{(x,y) \sim p_{\text{data}} \otimes p_{\text{data}}} \left[ \exp \left( -\frac{\|x^g - y^g\|^2}{2\sigma^2} \right) \right], \quad (11)$$

which retrieves the objective of the alignment and uniformity in Wang & Isola (2020). Specifically, in equation 11 the first expectation is the same as $\mathcal{L}_{\text{align}}$, and the second expectation is the same as $\mathcal{L}_{\text{uniform}}$ in Wang & Isola (2020) (up to the logarithmic transformation). Therefore, under our Assumption 3, we naturally decompose the objective for the KL into two: the first term leads to the alignment of positive pairs while the second term results in the uniformity of data representations. Compared to Theorem 1 of Wang & Isola (2020), our conclusion is based on the lower bound of $f$-MI instead of the InfoNCE loss, and does not rely on the limit of infinite negative samples. More importantly, besides the KL divergence we can extend our conclusion to various $f$-divergences.

**Alignment** Consider the objective in eq. (9). For the first term, since $f$ is convex, the derivative $f'$ is monotonic increasing. Therefore, in the ideal case, maximizing the first term would yield $x^g = y^g$ for all $(x, y) \sim p_{\text{pos}}$, *i.e.*, similar sample pairs should have aligned representations.

**Uniformity** In Wang & Isola (2020) it is proved that for the KL divergence minimizing the second expectation in equation 11 gives the uniform distribution, if we have *infinite samples*. Here we discuss the general $f$-divergences with *finite samples* by considering equation 10, *i.e.*, our empirical estimation of the objective. We list some common choices of $f$-divergences in Table 1, and a more detailed version can be found in Table 4 in Appendix A.1. We have the following theorem:

**Theorem 5 (uniformity).** *Suppose that the batch size $N$ satisfies $2 \leq N \leq d + 1$, with $d$ the dimension of the feature space. If the real function*

$$h(t) = f^* \circ f' \circ G_\sigma(t) \text{ is strictly convex on } [0, 4], \quad (12)$$

*then all minimizers of the second term of equation 10, i.e.,* $\sum_{i \neq j} f^* \circ f' \circ G_\sigma(\|x_i^g - x_j^g\|^2)$*, satisfy the following condition: the feature representations of all samples are distributed uniformly on the unit hypersphere* $\mathbb{S}^{d-1}$*. All $f$-divergences in Table 1 satisfy equation 12.*

Here by "distributed uniformly" we mean that the feature vectors form a regular simplex (see Figure 2), and thus the distances between all sample pairs are the same. It reflects our intuition that the feature embeddings are evenly distributed. Although minimizing the negative term gives uniformity, the positive term is also needed for aligning similar pairs, as we observe in our experiments in §5. Therefore, there is a *tradeoff* between alignment and uniformity. In Theorem 5 we assumed $N \leq d+1$, with $d$ as the feature dimension. This assumption is always satisfied in our experiments in §5. For instance, for CIFAR-10 experiments we choose $N = d = 512$.

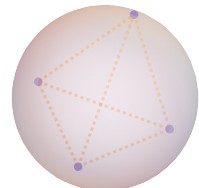

Figure 2: A regular simplex on a hypersphere.

Not all $f$-divergences lead to the property of uniformity. In Appendix A.1 we also listed some $f$-divergences that do not satisfy equation 12 and Theorem 5, such as the Reversed Kullback–Leibler (RKL) and the Neynman $\chi^2$ divergences. Experimentally, we found that these divergences generally result in feature collapse (*i.e.*, all feature vectors are the same) and thus poor performance in downstream applications. Compared to the existing literature (Nowozin et al., 2016; Acuna et al., 2021) where no suggestion is given for the choice of $f$-divergences theoretically, with Theorem 5 we provide guidance for choosing desirable $f$-divergences.

## 4 ESTIMATION OF $f$-MUTUAL INFORMATION

In §3.1 we have provided the empirical estimation of our objective in equation 10. However, it remains a question whether our estimation of $f$-mutual information is consistent. In this part we give an upper bound for the estimation error. Recall that our $f$-MICL objective is $i_f(X;Y)$ (equation 2) and its empirical estimation is $\hat{i}_f(X;Y)$ (equation 10), with $T(x,y) = f' \circ G_\sigma(\|x^g - y^g\|^2)$.

**Theorem 6** (**estimation error**). *Suppose that the function $T$ is taken from a function class $\mathcal{T}$ and define $\mathcal{T}_x$ as the function class of $T(x,\cdot)$ given some $x \in supp(p_{\text{data}})$. Denote $\mathfrak{R}_N^P$ to be the Rademacher complexity w.r.t. the distribution $P$ with $N$ i.i.d. drawn samples. Then for any $T \in \mathcal{T}$, the estimation error $|i_f(X;Y) - \hat{i}_f(X;Y)|$ is upper bounded with probability at least $1 - \delta$:*

$$2\mathfrak{R}_N^{P_{\text{pos}}}(\mathcal{T}) + 2\mu \left[ \mathbb{E}_{x \sim p_{\text{data}}} \mathfrak{R}_N^{p_{\text{data}}}(\mathcal{T}_x) + \frac{1}{N} \sum_{i=1}^N \mathfrak{R}_{N-1}^{p_{\text{data}}}(\mathcal{T}_{x_j}) \right] + (r_T + 2r_f)\sqrt{\frac{\log 6/\delta}{2(N-1)}}, \quad (13)$$

*with the constants $r_T = f'(\mu) - f'(\mu e^{-2/\sigma^2}), r_f = f^* \circ f'(\mu) - f^* \circ f'(\mu e^{-2/\sigma^2})$.*

Here the constant $\mu$ is from our Gaussian kernel in equation 7. Rademacher complexity evaluates the richness of a class of real-valued functions regarding a probability distribution, and its formal definition can be found in Koltchinskii (2001) (see also Definition 11 in Appendix B).

Note that the function class $\mathcal{T}$ depends on the class of the feature encoder $g$ and the $f$-divergence. The estimation error eq. (13) is composed of three parts: **(1)** the Rademacher complexity of the function class $\mathcal{T}$. In general, if $\mathcal{T}$ is richer then its Rademacher complexity is also larger. **(2)** the expected Rademacher complexity of the one-side function class $\mathcal{T}_x$ and its empirical estimation; and **(3)** an error term that decreases with the number of samples. Since the feature embeddings are usually neural networks, we can use existing theory (Bartlett et al., 2019) to give more detailed bounds for the Rademacher complexities of $\mathcal{T}$. Specifically, if the Vapnik–Chervonenkis (VC) dimension of $\mathcal{T}$ is finite, then our estimation error in eq. (13) goes to zero as $N \to \infty$ (Mohri et al., 2018).

**Approximation and estimation tradeoff** In order to minimize the estimation error in equation 13, we should choose a simpler function class $\mathcal{T}$ to reduce the Rademacher complexities. However, $\mathcal{T}$ should also be rich enough so that equation 3 can be satisfied, since our objective $i_f(X;Y)$ should approximate the $f$-mutual information $I_f(X;Y)$ if we choose the optimal $T$. Therefore, there is a natural tradeoff between approximation and estimation errors when we change the complexity of $\mathcal{T}$.

It is worth mentioning that our conclusion in Theorem 6 is theoretically non-trivial since our sample pairs are *non-i.i.d.*: although the individual samples are assumed to be i.i.d., the negative pairs are not independently drawn (*e.g.*, $(x_1, x_2)$ and $(x_1, x_3)$), which makes the derivation challenging.

## 5 EXPERIMENTS

We compare our framework with various frameworks on several popular vision and language datasets, and show the wide applicability of our method. In particular, our $f$-MICL gives state-of-the-art performance compared to popular baseline algorithms, such as SimCLR (van den Oord et al., 2018; Chen et al., 2020), MoCo (He et al., 2020), Uniformity (Wang & Isola, 2020), and RPC (Tsai et al., 2020). Additional experimental results can be seen in Appendix C.

Specifically, our results confirm the following:

- Our $f$-MICL encourages *alignment* between positive pairs, and encourages dissimilar sample pairs to be equally far apart and thus leads to *uniformity*.
- By replacing the cosine similarity with the Gaussian kernels, the performance is consistently better across a variety of $f$-divergences in our $f$-MICL framework.

### 5.1 EXPERIMENTAL SETTINGS

For vision tasks, we use SGD with momentum as our optimizer, and apply the cosine learning rate schedule (Loshchilov & Hutter, 2017). Regarding the neural network architecture, we employ ResNets (He et al., 2020) as the feature encoders. For the language dataset, we use the BERT models (Devlin et al., 2019) and Adam with weight decay as the optimizer (Loshchilov & Hutter, 2018). To achieve fair comparison, we keep the network architecture and optimization the same, while only changing the objective accordingly in our comparison. See Appendix C for more detailed settings.

### 5.2 COMPARISON WITH BENCHMARKS

We compare with several state-of-the-art benchmarks in Table 2. Note that SimCLR and RPC use the cosine similarity while we use the proposed Gaussian similarity.

**Vision task**  Our vision datasets include CIFAR-10, CIFAR-100 (Krizhevsky et al., 2009), STL-10 (Coates et al., 2011), TinyImageNet (Chrabaszcz et al., 2017) and ImageNet (Deng et al., 2009) for image classification. After learning a feature embedding, we evaluate the quality of representation using the test classification accuracies via a linear classifier. We observe from Table 2 that our proposed $f$-MICL consistently outperforms the benchmarks across all datasets. Specifically, we find that the JS divergence is superior in general, especially on the large datasets.

**Language task**  To show the efficacy of our $f$-MICL framework, we also conduct experiments on a natural language dataset: English Wikipedia (Gao et al., 2021). We follow the experimental setting of Gao et al. (2021), which applies SimCLR with BERT models (Devlin et al., 2019; Liu et al., 2019). The application task is semantic textual similarity (STS, Agirre et al. 2013) and we report the averaged the Spearman's correlation in Table 2 for comparison. We can see that our $f$-MICL performs better or on par with the benchmarks, especially for the KL divergence.

### 5.3 ABLATION STUDY

To develop a thorough understanding of $f$-MICL we also perform the ablation study regarding the choice of the similarity function, the design of the $f$-divergences, and the batch size. More ablation study can be found in Appendix C.

**Cosine similarity vs. Gaussian**  In Table 3 We compare the cosine and Gaussian similarities for different $f$-divergences on CIFAR-10. It can be seen that under our $f$-MICL framework the Gaussian similarity consistently outperforms the cosine similarity for various $f$-divergences. This agrees with our Theorem 2 and equation 8, and also implies the validity of Assumption 3.

**Non-satisfying $f$-divergences**  We have shown analytically that not all $f$-divergences are proper in our $f$-MICL framework (see the proof of Theorem 5, Appendix B). For example, in our experiments on CIFAR-10 and CIFAR-100, the RKL and Neyman $\chi^2$ do not perform well: the test classification accuracies on these two datasets are $10.00\%$ and $1.00\%$, respectively, which means that the classifier simply outputs the random guess. Moreover, we observe from Figure 3 that for the non-satisfying $f$-divergences such as the RKL, the features collapse to a constant. These observations emphasize the importance of choosing proper $f$-divergences and confirms our conclusion in Theorem 5.

Table 2: Test classification accuracy (%) on the vision datasets. For the Wikipedia dataset we evaluate the semantic textual similarity (STS) via the Spearman's correlation. For ImageNet we train for 100 epochs with batch size 256 due to computation limit. See Appendix C for detailed experimental settings.

| Dataset | Baselines | | | | $f$-MICL | | | |
|---|---|---|---|---|---|---|---|---|
| | MoCo | SimCLR | Uniformity | RPC | KL | JS | Pearson | VLC |
| CIFAR-10 | $90.30_{\pm0.19}$ | $89.71_{\pm0.37}$ | $90.41_{\pm0.26}$ | $90.39_{\pm0.25}$ | $\mathbf{90.61_{\pm0.47}}$ | $89.66_{\pm0.28}$ | $89.35_{\pm0.52}$ | $89.13_{\pm0.33}$ |
| CIFAR-100 | $62.77_{\pm0.17}$ | $62.75_{\pm0.45}$ | $62.51_{\pm0.36}$ | $62.66_{\pm0.39}$ | $63.00_{\pm0.44}$ | $\mathbf{63.11_{\pm0.33}}$ | $61.69_{\pm0.57}$ | $61.19_{\pm0.29}$ |
| STL-10 | $83.69_{\pm0.22}$ | $82.97_{\pm0.32}$ | $84.44_{\pm0.19}$ | $82.41_{\pm0.14}$ | $85.33_{\pm0.39}$ | $\mathbf{85.94_{\pm0.17}}$ | $82.64_{\pm0.37}$ | $83.27_{\pm0.72}$ |
| TinyImageNet | $35.72_{\pm0.17}$ | $30.56_{\pm0.28}$ | $41.20_{\pm0.19}$ | $34.95_{\pm0.25}$ | $39.46_{\pm0.20}$ | $\mathbf{42.98_{\pm0.18}}$ | $38.45_{\pm0.54}$ | $38.65_{\pm0.45}$ |
| ImageNet | $58.59$ | $57.66$ | $59.12$ | $56.11$ | $58.91$ | $\mathbf{61.11}$ | $55.33$ | $54.26$ |
| Wikipedia | $77.88_{\pm0.15}$ | $77.40_{\pm0.12}$ | $77.95_{\pm0.08}$ | $68.32_{\pm0.23}$ | $\mathbf{78.02_{\pm0.13}}$ | $76.76_{\pm0.09}$ | $77.59_{\pm0.12}$ | $55.07_{\pm0.13}$ |

Table 3: Comparison between the cosine similarity and our Gaussian similarity on CIFAR-10 using the test classification accuracy (%).

| Similarity | KL | JS | Pearson | SH | Tsallis | VLC |
|---|---|---|---|---|---|---|
| Cosine | 88.95 | 87.06 | 87.79 | 87.06 | 88.55 | 10.00 |
| Gaussian | **89.34** | **89.12** | **89.44** | **88.13** | **89.18** | **89.15** |

**Sensitivity to batch size** We study the sensitivity to the batch size of our $f$-MICL framework on CIFAR-10. On the right panel of Figure 3, we evaluate the classification accuracy by varying the batch size for different $f$-divergences and SimCLR. We can see that for all different batch sizes and with proper choice of $f$-divergences, our performance is always better than SimCLR. In other words, we require less negative samples to achieve the same performance.

## 5.4 UNIFORMITY TEST

To check the uniformity of feature vectors (Theorem 5) we plot the pairwise distance $\|x_i^g - x_j^g\|$ of the feature representations within the same batch on CIFAR-10 and CIFAR-100. We compute the distances between the normalized features of every pair from a random batch, and then sort the pairs with the increasing order. From Figure 3 we can see that $f$-MICL gives nearly uniform distances for dissimilar pairs (orange regions) on both datasets with various proper $f$-divergences. In contrast, a random initialized model gives a less uniform distribution for dissimilar pairs. Besides, for $f$-MICL we observe small pairwise distances for similar pairs (green regions). On CIFAR-100 there are less similar pairs compared to CIFAR-10 as there are more classes.

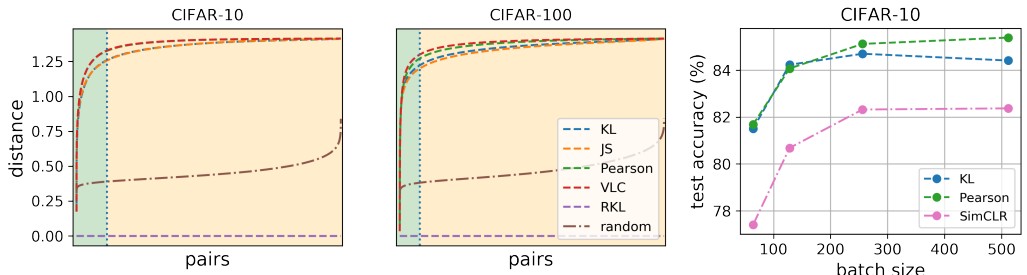

Figure 3: **(left and middle)** Distances between pairs of normalized features within a batch. **Green region:** similar pairs. **Orange region:** dissimilar pairs. $f$-MICL gives nearly uniform distances for dissimilar pairs for the $f$-divergences in Table 1. For non-satisfying $f$-divergences such as the RKL, the features collapse to a constant and thus the distances are zero. **(right)** The test classification accuracy v.s. the batch size after training 200 epochs for all algorithms.

## 6 RELATED WORKS

**Contrastive learning**  Self-supervised contrastive learning learns representations by encouraging the contrastiveness between positive pairs and negative pairs. Recently it has been shown analytically that improving the contrastiveness can benefit the downstream applications (Saunshi et al., 2019; Tosh et al., 2021). For popular contrastive learning methods such as Contrastive Predictive Coding (CPC) (van den Oord et al., 2018), SimCLR (Chen et al., 2020), and MoCo (He et al., 2020), their loss functions can be interpreted as a lower bound of mutual information, which is essentially the KL divergence between the joint distribution $p_{\text{pos}}$ and the product of margin distributions $p_{\text{data}} \otimes p_{\text{data}}$. Besides the KL divergence, other statistical divergences or distances have been individually studied under the context of contrastive learning, *e.g.*, the Wasserstein distance (Ozair et al., 2019), Pearson $\chi^2$ divergence (Tsai et al., 2021), and Jensen–Shannon divergence (Hjelm et al., 2018). Our paper differs from Hjelm et al. (2018); Tsai et al. (2021) in the following ways:

- Instead of treating $f$-divergences as estimators of mutual information, we directly consider $f$-*mutual information*, which builds solid foundation for our objective.
- Instead of using the default cosine similarity, we challenge this routine and construct the optimal design with the Gaussian kernel, which consistently performs better (Table 3).
- We show that optimizing the $f$-MICL objective of the contrastive learning yields alignment and uniformity for proper $f$-divergences with *finite samples*.

**Uniformity**  Our work is inspired by Wang & Isola (2020), which shows the alignment and uniformity in contrastive learning. However, Wang & Isola (2020) focuses on the InfoNCE loss and the loss function $\mathcal{L}_{\text{uniform}}$ they consider is somewhat *ad hoc*. In contrast, our $f$-MICL objective is directly built from the $f$-mutual information theory. Another difference is that Wang & Isola (2020) proves uniformity in the limit of infinite samples, while our Theorem 5 only needs finite samples.

**$f$-divergences** have been widely used in, *e.g.*, generative models (Nowozin et al., 2016), variational inference (Wan et al., 2020), and domain adaptation (Acuna et al., 2021), for measuring the discrepancy of two distributions. Compared to the existing work such as $f$-GAN (Nowozin et al., 2016) which evaluates the similarity of two marginal distributions (*i.e.*, the real and the modeled data distributions), our $f$-MICL objective measures the similarity between the joint distribution and the product of data distributions. Furthermore, while $f$-GAN aims to learn a generative model and thus *minimizes* the $f$-divergence, our $f$-MICL intends to encourage the contrastiveness and *maximizes* the $f$-divergence. Finally, we provide a theoretical criterion for choosing proper $f$-divergences.

**Metric learning**  Our work is closely related to metric learning (Kaya & Bilge, 2019; Suárez-Díaz et al., 2018), which aims to learn a distance metric bringing similar objects closer and distancing dissimilar objects further. Distance metrics that are designed heuristically or independently of the problem may not lead to a satisfactory performance in applications. In contrastive learning, a predefined similarity metric, *e.g.*, the cosine similarity (Chen et al., 2020; He et al., 2020) or a bilinear function (van den Oord et al., 2018; Tian et al., 2020a; Hénaff et al., 2020) is commonly employed to measure the similarity of sample pairs. Comparably, we have derived an optimal similarity function assuming that the joint positive distribution is proportional to a Gaussian kernel.

## 7 CONCLUSION AND FUTURE WORK

In this work we proposed $f$-MICL, a novel contrastive learning framework with a generalization of mutual information, called $f$-*mutual information* ($f$-MI). Our objective corresponds to the variational bound of $f$-divergences. By making the assumption that the joint feature distribution is proportional to a Gaussian kernel, we naturally characterized the properties of the alignment and uniformity. Our $f$-MI based objective can be well estimated with finite samples. Experimentally, we showed the efficacy of $f$-MICL across a wide array of datasets and the advantage of using the Gaussian similarity. Our results imply the following: **(1)** even though mutual information is widely used in contrastive learning, generalization to $f$-MI can bring us better performance; **(2)** the cosine similarity, the *de facto* option in contrastive learning, can be replaced with more effective similarity functions such as the Gaussian similarity. As our work shows interesting connection between contrastive learning and kernel methods, it would be promising to explore the usage of more RBF kernels in contrastive learning.

## REPRODUCIBILITY STATEMENT

Our novel $f$-MICL objectives are implemented in anonymous downloadable source code as in the supplementary. For theoretical results, clear explanations of any assumptions and complete proofs are included in Appendix A and Appendix B. For detailed experimental settings and complete experimental results, the complete descriptions can be found in Appendix C.

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

Table 4: A summary of common $f$-divergences. KL: Kullback–Leibler; JS: Jensen–Shannon; SH: Squared Hellinger. For JS, we define $\varphi(u) = -(u+1)\log\frac{1+u}{2} + u\log u$. For Pearson $\chi^2$, we take $f^*(t) = -1$ if $t \leq -2$. For Jeffrey, $\widehat{W} = W + W^{-1}$ and $W(\cdot)$ is the Lambert-$W$ product log function. The Tsallis-$\alpha$ divergence is defined in Tsallis (1988) and we have $\alpha > 1$ for $f$-divergences. We ignore constant addition $-1/(\alpha-1)$ because it does not change the optimization problem. The Vincze–Le Cam divergence can be found in (p.47, Le Cam, 2012) which is closely related to $\chi^2$ and Hellinger divergences. For the Vincze–Le Cam divergence we require $-3 < t < 1$ and $f^*(t) = -1$ if $t \leq -3$.

| **Divergence** | $f(u)$ | $f^*(t)$ | $f'(u)$ | $f^* \circ f'(u)$ |
|---|---|---|---|---|
| KL | $u\log u$ | $\exp(t-1)$ | $\log u + 1$ | $u$ |
| Reverse KL | $-\log u$ | $-1 - \log(-t)$ | $-1/u$ | $\log u - 1$ |
| JS | $\varphi(u)$ | $-\log(2 - e^t)$ | $\log 2 + \log\frac{u}{1+u}$ | $-\log 2 + \log(1+u)$ |
| Pearson $\chi^2$ | $(u-1)^2$ | $t^2/4 + t$ | $2(u-1)$ | $u^2 - 1$ |
| SH | $(\sqrt{u}-1)^2$ | $\frac{t}{1-t}$ | $1 - u^{-1/2}$ | $u^{1/2} - 1$ |
| Neyman $\chi^2$ | $\frac{(1-u)^2}{u}$ | $2 - 2\sqrt{1-t}$ | $1 - u^{-2}$ | $2 - 2u^{-1}$ |
| Jeffrey | $(u-1)\log u$ | $\widehat{W}(e^{1-t}) + t - 2$ | $1 - u^{-1} + \log u$ | $\widehat{W}(e^{1/u}/u) + \log u - \frac{1+u}{u}$ |
| Tsallis $\alpha$ | $u^\alpha/(\alpha-1)$ | $((\alpha-1)t/\alpha)^{\alpha/(\alpha-1)}$ | $\frac{\alpha u^{\alpha-1}}{\alpha-1}$ | $u^\alpha$ |
| Vincze–Le Cam | $\frac{(u-1)^2}{u+1}$ | $4 - t - 4\sqrt{1-t}$ | $\frac{(u-1)(u+3)}{(u+1)^2}$ | $3 - \frac{4}{u+1}$ |

## A    ADDITIONAL THEORETICAL RESULTS

In this appendix we provide additional theoretical results, including additional $f$-divergences and the theory for weighting parameters.

### A.1    ADDITIONAL $f$-DIVERGENCES

We expand Table 1 and give more examples of $f$-divergences in Table 4. As we will see in the proof of Theorem 5, Table 1 gives a special class of $f$-divergences that guarantees uniformity. A detailed description of $f$-divergences can be found in e.g. Sason & Verdú (2016).

### A.2    WEIGHTING PARAMETERS

In Algorithm 1 we added a weighting parameter $\alpha$ to balance the alignment and uniformity. We prove that even after adding this parameter we are still maximizing the $f$-mutual information, although with respect to a different $f$.

**Proposition 7 (weighting parameter).** *Given $\alpha > 0$ and a closed convex function $f : \mathbb{R}_+ \to \mathbb{R}$ such that $f(1) = 0$, define $f_\alpha : \alpha\,dom\,f \to \mathbb{R}$ with $f_\alpha(x) = \alpha f(x/\alpha) - \alpha f(1/\alpha)$ for any $x \in dom\,f$. Then $I_{f_\alpha}$ is still a valid $f$-mutual information (see Definition 1). Besides, by replacing $f$ with $f_\alpha$ in equation 9 we have the following optimization problem:*

$$\sup_{g \in \mathcal{G}} \mathbb{E}_{(x,y)\sim p_{\text{pos}}} \left[ f'\left(G_\sigma(\|x^g - y^g\|^2)/\alpha\right) \right] - \alpha \mathbb{E}_{(x,y)\sim p_{\text{data}} \otimes p_{\text{data}}} \left[ f^* \circ f'\left(G_\sigma(\|x^g - y^g\|^2)/\alpha\right) \right],$$

*where $G_\sigma(\|x^g - y^g\|^2) = \mu\exp\left(-\frac{\|x^g - y^g\|^2}{2\sigma^2}\right)$ is the Gaussian kernel.*

Note that $\alpha\,dom\,f$ means the scalar multiplication of a set which is applied element-wisely. According to Definition 1, $f_\alpha$ is also a valid $f$-divergence. This proposition tells us that rescaling the second term with factor $\alpha$ is equivalent to changing the function $f$ to another convex function $f_\alpha$. The transformation from $f$ to $\alpha f(x/\alpha)$ is also known as right scalar multiplication (e.g. Chapter X, Urruty & Lemaréchal, 1993). Let us now move on to our proof:

*Proof.* By definition we know that $f_\alpha$ is convex and closed with $f_\alpha(1) = 0$, and thus $I_{f_\alpha}$ is a valid $f$-mutual information according to Definition 1. Moreover, we have $f'_\alpha(x) = f'(x/\alpha)$ for any

$x \in \alpha \operatorname{dom} f$ and

$$
\begin{aligned}
f_\alpha^*(t) &= \sup_{x \in \operatorname{dom} f_\alpha} xt - f_\alpha(x) \\
&= \sup_{x \in \alpha \operatorname{dom} f} xt - \alpha f(x/\alpha) + \alpha f(1/\alpha) \\
&= \sup_{x/\alpha \in \operatorname{dom} f} (x/\alpha) \cdot (\alpha t) - \alpha f(x/\alpha) + \alpha f(1/\alpha) \\
&= \alpha \sup_{x/\alpha \in \operatorname{dom} f} ((x/\alpha) \cdot t - f(x/\alpha)) + \alpha f(1/\alpha) \\
&= \alpha f^*(t) + \alpha f(1/\alpha),
\end{aligned}
\tag{A.1}
$$

where in the last line we used the definition of $f^*(t)$. Plugging $f_\alpha'$ and $f_\alpha^*$ into equation 9 yields the desired result. $\square$

### A.3 Uniform distributions

In the following we prove that under Assumption 3 the marginal feature distribution $p_{\text{data}}^g$ is uniform.

**Proposition 8.** *Under Assumption 3, the marginal feature distribution $p_{\text{data}}^g$ is uniform on the hypersphere $\mathbb{S}^{d-1}$, with $d$ the dimension of the feature space.*

*Proof.* Under Assumption 3 we have:

$$
p_g(x^g, y^g) = C_0 \exp\left( -\frac{\|x^g - y^g\|^2}{2\sigma^2} \right),
\tag{A.2}
$$

where $C_0$ is a normalizing constant. Integrating $y^g$ we have the marginal distribution for $x^g$:

$$
p_g(x^g) = \int_{\mathbb{S}^{d-1}} C_0 \exp\left( -\frac{\|x^g - y^g\|^2}{2\sigma^2} \right) dy^g.
\tag{A.3}
$$

It suffices to show that $p_g(x_1^g) = p_g(x_2^g)$ for any $x_1^g, x_2^g \in \mathbb{S}^{d-1}$. Suppose $Q$ is the orthogonal matrix such that:

$$
Qx_1^g = x_2^g.
\tag{A.4}
$$

Such a matrix $Q$ always exists and constructing $Q$ is not difficult. For example, assume that $\{x_1^g, x_2^g\}$ span a plane with an orthonormal basis $e_1, e_2$, and

$$
\begin{aligned}
x_1^g &= \cos\theta_1 \cdot e_1 + \sin\theta_1 \cdot e_2, \\
x_2^g &= \cos\theta_2 \cdot e_1 + \sin\theta_2 \cdot e_2,
\end{aligned}
\tag{A.5}
$$

then $Q$ can take the following form:

$$
Q = [e_1 \; e_2] \begin{bmatrix} \cos(\theta_2 - \theta_1) & -\sin(\theta_2 - \theta_1) \\ \sin(\theta_2 - \theta_1) & \cos(\theta_2 - \theta_1) \end{bmatrix} \begin{bmatrix} e_1^\top \\ e_2^\top \end{bmatrix}
\tag{A.6}
$$

such that $Q$ is orthogonal and satisfies $Qx_1^g = x_2^g$. Hence we have:

$$
\begin{aligned}
p_g(x_1^g) &= \int_{\mathbb{S}^{d-1}} C_0 \exp\left( -\frac{\|x_1^g - y^g\|^2}{2\sigma^2} \right) dy^g \\
&= \int_{\mathbb{S}^{d-1}} C_0 \exp\left( -\frac{\|Q^\top x_2^g - y^g\|^2}{2\sigma^2} \right) dy^g \\
&= \int_{\mathbb{S}^{d-1}} C_0 \exp\left( -\frac{\|Q^\top x_2^g - Q^\top z^g\|^2}{2\sigma^2} \right) d(Q^\top z^g) \\
&= \int_{\mathbb{S}^{d-1}} C_0 \exp\left( -\frac{\|x_2^g - z^g\|^2}{2\sigma^2} \right) dz^g \\
&= p_g(x_2^g),
\end{aligned}
\tag{A.7}
$$

where in the second line we used equation A.4; in the third line we made the transformation $y^g = Q^\top z^g$ with $z^g$ a unit vector; in the fourth line we used the fact that applying a orthogonal matrix does not change the norm and that the corresponding Jacobian determinant is one.

In fact, the proof above can be generalized from the Gaussian kernel to any radial basis functions, by replacing the Gaussian kernel with $\varphi(\|x^g - y^g\|^2)$, and repeating the same proof. Here $\varphi$ can be any function such that the integral $\int_{\mathbb{S}^{d-1}} \varphi(\|x^g - y^g\|^2) dy^g$ is finite. $\square$

# B  PROOFS

**Lemma 2** (*e.g.*, Nguyen et al. 2010, Lemma 1). *Suppose $f$ is differentiable, and the encoder function $g$ is fixed. The similarity function*

$$k^*(x^g, y^g) = f'\left(\frac{p_g(x^g, y^g)}{p_g(x^g)p_g(y^g)}\right) \tag{6}$$

*maximizes $i_f(X;Y)$ in eq. (5) as long as it is contained in the function class $\mathcal{K}$.*

*Proof.* From Definition 1, we are computing the following supremum:

$$\sup_{g,k} \int \left(\frac{p_g(x^g, y^g)}{p_g(x^g)p_g(y^g)}k(x^g, y^g) - f^* \circ k(x^g, y^g)\right) dp_{\texttt{data}}^g \otimes p_{\texttt{data}}^g. \tag{B.1}$$

Suppose $k$ is unconstrained and we fix $g$. The optimal solution should satisfy:

$$\frac{p_g(x^g, y^g)}{p_g(x^g)p_g(y^g)} \in (\partial f^*)(k^*(x^g, y^g)), \tag{B.2}$$

almost surely for $(x, y) \sim p_{\texttt{data}} \otimes p_{\texttt{data}}$. From (3.11) of Rockafellar (1966) this is equivalent to:

$$k^*(x^g, y^g) \in \partial f\left(\frac{p_g(x^g, y^g)}{p_g(x^g)p_g(y^g)}\right). \tag{B.3}$$

If $f$ is differentiable, then for any $u \in \text{dom } f$, $\partial f(u) = \{f'(u)\}$ is a singleton. $\square$

**Theorem 4** (**Gaussian similarity**). *Under Assumption 3 with Gaussian kernels and the same settings as Lemma 2, the optimal similarity function $k^*$ satisfies that for any $x^g, y^g \in \mathbb{S}^{d-1}$:*

$$k^*(x^g, y^g) = f'(CG_\sigma(\|x^g - y^g\|^2)), \tag{8}$$

*where $d$ is the feature dimension and $C$ is an absolute constant.*

*Proof.* Simply combine Proposition 8 with Lemma 2. $\square$

**Theorem 5** (**uniformity**). *Suppose that the batch size $N$ satisfies $2 \leq N \leq d+1$, with $d$ the dimension of the feature space. If the real function*

$$h(t) = f^* \circ f' \circ G_\sigma(t) \text{ is strictly convex on } [0, 4], \tag{12}$$

*then all minimizers of the second term of equation 10, i.e., $\sum_{i \neq j} f^* \circ f' \circ G_\sigma(\|x_i^g - x_j^g\|^2)$, satisfy the following condition: the feature representations of all samples are distributed uniformly on the unit hypersphere $\mathbb{S}^{d-1}$. All $f$-divergences in Table 1 satisfy equation 12.*

*Proof.* From the definition of $h$ it is clear that $h$ is decreasing since $f^*$ and $f'$ are both monotonically increasing white $G_\sigma$ is decreasing. Using $h$ we rewrite the second term of equation 10 as

$$\min_{x_1^g, \dots, x_N^g \in \mathbb{S}^{d-1}} \sum_{i,j} h(\|x_i^g - x_j^g\|^2). \tag{B.4}$$

When $N \in [2, d+1]$, there exists a neat characterization of the minimizers, see e.g. Borodachov et al. (2019, Theorem 2.4.1). We include the proof below for completeness.

Apply Jensen's inequality, we have:

$$
\begin{aligned}
\frac{1}{N^2} \sum_{i,j} h(\|x_i - x_j\|^2) &\geq h\left(\frac{1}{N^2} \sum_{i,j} \|x_i - x_j\|^2\right) \\
&= h\left(\frac{1}{N^2} \sum_{i,j} \|x_i - x_j\|^2\right) \\
&= h\left(\frac{1}{N^2} \sum_{i,j} (2 - 2x_i \cdot x_j)\right) \\
&= h\left(2\left(1 - \left\|\frac{1}{N}\sum_{i=1}^{N} x_i\right\|^2\right)\right) \\
&\geq h(2),
\end{aligned}
\tag{B.5}
$$

where in the first line we used Jensen's inequality; in the third line we used $\|x_i\| = \|x_j\| = 1$ for any $i, j \in [N]$; in the last line we note that $\|\sum_{i=1}^{N} x_i\| \geq 0$ and $h$ is a decreasing function. When $h$ is strictly convex and decreasing, it is in fact strictly decreasing, and hence the two inequalities above can be attained iff

$$
\bar{x} := \frac{1}{N}\sum_i x_i = \mathbf{0}, \quad \text{and } \|x_i - x_j\|^2 \equiv c \text{ for all } i \neq j,
\tag{B.6}
$$

namely that $\{x_1, \ldots, x_N\}$ form a regular simplex with its center at the origin. We remark that when $h$ is merely convex, points forming a centered regular simplex may form a strict subset of the minimizers.

To see the necessity of $N \leq d + 1$, let us note that

$$
x_i^\top x_j = \begin{cases} 1, & i = j \\ -\frac{1}{N-1}, & i \neq j \end{cases},
\tag{B.7}
$$

since

$$
\sum_{ij} \|x_i - x_j\|^2 = 2N^2 = N(N-1)c \implies c = \frac{2N}{N-1} = 2 + \frac{2}{N-1}.
\tag{B.8}
$$

Performing simple Gaussian elimination we note that the matrix $X^\top X$ has rank $N - 1$ where $X = [x_1, \ldots, x_N] \in \mathbb{R}^{d \times N}$. Therefore, we must have $N - 1 \leq d$.

Lastly, we need to show when $h$ is a (strictly) convex function, which may not always be true depending on the $f$-divergences. We give the following characterization (we ignore the constants $\mu$ and $2\sigma^2$ in equation 7 as they do not affect convexity):

- $h$ strictly convex: $h_{\text{KL}}(t) = e^{-t}$, $h_{\text{JS}}(t) = \log(1 + e^{-t}) - \log 2$, $h_{\text{Pearson}}(t) = e^{-2t} - 1$, $h_{\text{SH}}(t) = e^{-t/2} - 1$, $h_{\text{Tsallis}}(t) = e^{-\alpha t}$, $h_{\text{VLC}} = 3 - \frac{4}{1+e^{-t}}$;

- $h$ convex but not strictly convex: $h_{\text{RKL}}(t) = -t - 1$ (RKL stands for Reversed Kullback–Leibler, see Appendix A.1);

- $h$ concave: $h_{\text{Neyman}}(t) = 2 - 2e^t$ (Neyman stands for Neyman $\chi^2$, see Appendix A.1).

Only for the last case we do not have the guarantee that the minimizing configurations could form a regular simplex. For RKL, in fact, any configuration that centers at the origin suffices since $h$ is a linear function. $\qquad\square$

**Theorem 6** (**estimation error**). *Suppose that the function $T$ is taken from a function class $\mathcal{T}$ and define $\mathcal{T}_x$ as the function class of $T(x, \cdot)$ given some $x \in supp(p_{\text{data}})$. Denote $\mathfrak{R}_N^P$ to be the*

*Rademacher complexity w.r.t. the distribution $P$ with $N$ i.i.d. drawn samples. Then for any $T \in \mathcal{T}$, the estimation error $|i_f(X;Y) - \hat{i}_f(X;Y)|$ is upper bounded with probability at least $1 - \delta$:*

$$2\mathfrak{R}_N^{p_{\text{pos}}}(\mathcal{T}) + 2\mu \left[ \mathbb{E}_{x \sim p_{\text{data}}} \mathfrak{R}_N^{p_{\text{data}}}(\mathcal{T}_x) + \frac{1}{N} \sum_{i=1}^{N} \mathfrak{R}_{N-1}^{p_{\text{data}}}(\mathcal{T}_{x_j}) \right] + (r_T + 2r_f)\sqrt{\frac{\log 6/\delta}{2(N-1)}}, \quad (13)$$

*with the constants $r_T = f'(\mu) - f'(\mu e^{-2/\sigma^2}), r_f = f^* \circ f'(\mu) - f^* \circ f'(\mu e^{-2/\sigma^2})$.*

*Proof.* Our proof uses the following notion of Rademacher complexity:

**Definition 11** (Rademacher complexity, Koltchinskii (2001))**.** *Let $\mathcal{F}$ be a family of functions from a subset of Euclidean space $\mathcal{W}$ to $[a, b]$ and $S = (w_1, \ldots, w_m)$ a fixed sample of size $m$ with elements in $\mathcal{W}$. The empirical Rademacher complexity of $\mathcal{F}$ w.r.t. the sample set $S$ is defined as:*

$$\hat{\mathfrak{R}}_S(\mathcal{F}) = \mathbb{E}_{\sigma_i} \left[ \sup_{f \in \mathcal{F}} \frac{1}{m} \sum_{i=1}^{m} \sigma_i f(w_i) \right], \quad (B.9)$$

*where $\sigma_i$'s are independent uniform random variables taking values $\{-1, +1\}$. Let $P$ be the underlying distribution of samples. The Rademacher complexity $\mathfrak{R}_m(\mathcal{F})$ of function class $\mathcal{F}$ is the expectation over sample sets:*

$$\mathfrak{R}_m^P(\mathcal{F}) = \mathbb{E}_{S \sim P^m}[\hat{\mathfrak{R}}_S(\mathcal{F})]. \quad (B.10)$$

With the definition of Rademacher complexity we have the following lemma:

**Lemma 12.** *Let $\mathcal{F}$ be a family of functions from a subset of Euclidean space $\mathcal{W}$ to $[a, b]$ with $b > a$. Then for any $0 < \delta < 1$ with probability at least $1 - \delta$, the following holds for any $f \in \mathcal{F}$:*

$$|\mathbb{E}_{w \sim P_w}[f(w)] - \frac{1}{m} \sum_{i=1}^{m} f(w_i)| \leq 2\mathfrak{R}_m^{P_w}(\mathcal{F}) + (b - a)\sqrt{\frac{\log(2/\delta)}{2m}}, \quad (B.11)$$

*with $w_i$ i.i.d. examples from any distribution $P_w$.*

*Proof.* Our proof follows from Lemma 11 of Zhang et al. (2021). For any function $f \in \mathcal{F}$ we can construct $\tilde{f} = (f - a)/(b - a)$ such that $\tilde{f} : \mathcal{W} \to [0, 1]$. The rest follows from the linearity of Rademacher complexity. $\square$

Let us first recall our definition equation 4: $T(x, y) := k(g(x), g(y))$. From the definitions of $i_f(X;Y)$ and $\hat{i}_f(X;Y)$ we write the following:

$$|i_f(X;Y) - \hat{i}_f(X;Y)| = \left| \mathbb{E}_{(x,z) \sim p_{\text{pos}}}[T(x, z)] - \mathbb{E}_{x \sim p_{\text{data}}} \mathbb{E}_{y \sim p_{\text{data}}}[f^*(T(x, y))] - \right.$$

$$\left. - \left( \frac{1}{N} \sum_{i=1}^{N} T(x_i, z_i) - \frac{1}{N(N-1)} \sum_{i \neq j} f^*(T(x_i, x_j)) \right) \right|$$

$$\leq \left| \mathbb{E}_{(x,z) \sim p_{\text{pos}}}[T(x, z)] - \frac{1}{N} \sum_{i=1}^{N} T(x_i, z_i) \right| +$$

$$+ \left| \mathbb{E}_{x \sim p_{\text{data}}} \mathbb{E}_{y \sim p_{\text{data}}}[f^*(T(x, y))] - \frac{1}{N(N-1)} \sum_{i \neq j} f^*(T(x_i, x_j)) \right|,$$

$$\text{(B.12)}$$

where we used the triangle inequality. With Lemma 12, we can upper bound the first term of equation B.12 as:

$$2\mathfrak{R}_N^{p_{\text{pos}}}(\mathcal{T}) + (b_T - a_T)\sqrt{\frac{\log(2/\delta)}{2N}}, \text{ w.p. } 1 - \delta, \quad (B.13)$$

where $[a_T, b_T]$ is the range of $T$. Now we upper bound the second term of equation B.12. Note that we cannot apply Lemma 12 to equation B.12 directly since $(x_i, x_j)$'s are not i.i.d. samples. For instance, the pairs $(x_1, x_2)$ and $(x_1, x_3)$ are not independent since they share the same $x_1$. However, all $x_i$'s are i.i.d. sampled. We can upper bound the second term of equation B.12 as:

$$\left| \mathbb{E}_{x \sim p_{\text{data}}} \mathbb{E}_{y \sim p_{\text{data}}} f^*(T(x,y)) - \frac{1}{N(N-1)} \sum_{i \neq j} f^*(T(x_i, x_j)) \right| =$$

$$\left| \mathbb{E}_{x \sim p_{\text{data}}} \mathbb{E}_{y \sim p_{\text{data}}} f^*(T(x,y)) - \mathbb{E}_{x \sim p_{\text{data}}} \frac{1}{N} \sum_{j=1}^{N} f^*(T(x, x_j)) + \right.$$

$$\left. + \mathbb{E}_{x \sim p_{\text{data}}} \frac{1}{N} \sum_{j=1}^{N} f^*(T(x, x_j)) - \frac{1}{N(N-1)} \sum_{i \neq j} f^*(T(x_i, x_j)) \right| \leq$$

$$\left| \mathbb{E}_{x \sim p_{\text{data}}} \mathbb{E}_{y \sim p_{\text{data}}} f^*(T(x,y)) - \mathbb{E}_{x \sim p_{\text{data}}} \frac{1}{N} \sum_{j=1}^{N} f^*(T(x, x_j)) \right| +$$

$$+ \left| \mathbb{E}_{x \sim p_{\text{data}}} \frac{1}{N} \sum_{j=1}^{N} f^*(T(x, x_j)) - \frac{1}{N(N-1)} \sum_{i \neq j} f^*(T(x_i, x_j)) \right|. \quad (B.14)$$

The first term of equation B.14 can be upper bounded as:

$$\left| \mathbb{E}_{x \sim p_{\text{data}}} \left[ \mathbb{E}_{y \sim p_{\text{data}}} f^*(T(x,y)) - \sum_{j=1}^{N} f^*(T(x, x_j)) \right] \right|$$

$$\leq 2 \mathbb{E}_{x \sim p_{\text{data}}} \mathfrak{R}_N^{p_{\text{data}}}(f^* \circ \mathcal{T}_x) + (b_f - a_f) \sqrt{\frac{\log(2/\delta)}{2N}}$$

$$\leq 2 L_f \mathbb{E}_{x \sim p_{\text{data}}} \mathfrak{R}_N^{p_{\text{data}}}(\mathcal{T}_x) + (b_f - a_f) \sqrt{\frac{\log(2/\delta)}{2N}} \quad (B.15)$$

with probability at least $1 - \delta$, where we used Lemma 12 in the first line and Talagrand's lemma (*e.g.*, Lemma 5.7 of Mohri et al. (2018)) in the second line. We denote $L_f$ as the Lipschitz constant of $f^*$ and $[a_f, b_f]$ as the range of $f^* \circ T$. Similarly, we can upper bound the second term of eq. (B.14) as:

$$\frac{L_f}{N} \sum_{j=1}^{N} \mathfrak{R}_{N-1}^{p_{\text{data}}}(\mathcal{T}_{x_j}) + (b_f - a_f) \sqrt{\frac{\log(2/\delta)}{2(N-1)}}, \quad (B.16)$$

with probability at least $1 - \delta$.

**Absolute constants**   Let us compute the constants $a_f, b_f, a_T, b_T, L_f$. First note that our feature embeddings are normalized. For any $x^g, y^g \in \mathbb{S}^{d-1}$ we have:

$$0 \leq \|x^g - y^g\|^2 \leq 2(\|x^g\|^2 + \|y^g\|^2) = 4. \quad (B.17)$$

From the expression of Gaussian similarity equation 7 and the monotonicity of $f'$ and $f^*$, we have:

$$b_T = f'(\mu), a_T = f'(\mu e^{-2/\sigma^2}), b_f = f^* \circ f'(\mu), a_f = f^* \circ f'(\mu e^{-2/\sigma^2}). \quad (B.18)$$

For simplicity we will write $r_T = b_T - a_T$ and $r_f = b_f - a_f$. We can also compute the Lipschitz constant of $f^*$, which is bounded by the norm of the gradient:

$$L_f \leq \sup_{x^g, y^g \in \mathbb{S}^{d-1}} (f^*)' \circ f' \circ G_\sigma(\|x^g - y^g\|^2) = \sup_{x^g, y^g \in \mathbb{S}^{d-1}} G_\sigma(\|x^g - y^g\|^2) = \mu, \quad (B.19)$$

where we used the equality $(f^*)' \circ f' = \texttt{id}$ (Rockafellar, 1966), with $\texttt{id}$ the identity function.

Combining equation B.13, equation B.15 and equation B.16 we can use the union bound to finish the proof. $\qquad \square$

## C  ADDITIONAL EXPERIMENTAL RESULTS

We present additional experiment details in this appendix, to further support our experiments in the main paper.

### C.1  IMPLEMENTATION DETAILS

In this paper, we follow the implementations in SimCLR (`https://github.com/sthalles/SimCLR`). For vision tasks, we use ResNet (He et al., 2016) as the feature encoder, and we adopt the similar procedure of SimCLR for sampling. For the language dataset, we follow the exact experimental setting of Gao et al. (2021) and only change the objective. Our experimental settings are detailed below:

- Hardware and package: We train on a GPU cluster with `NVIDIA T4` and `P100`. The platform we use is `pytorch`. Specifically, the pairwise summation can be easily implemented using `torch.nn.functional.pdist` from `pytorch`.

- Datasets: the datasets we consider include CIFAR-10, CIFAR-100 (Krizhevsky et al., 2009), STL-10 (Coates et al., 2011), TinyImageNet (Chrabaszcz et al., 2017), ImageNet (Deng et al., 2009) and English Wikipedia (Gao et al., 2021).

- Augmentation method: For each sample in a dataset we create a sample pair, a.k.a. positive pair, using two different augmentation functions. For image samples, we choose the augmentation functions to be the standard ones in contrastive learning, e.g., in Chen et al. (2020) and He et al. (2020). The augmentation is a composition of random flipping, cropping, color jittering and gray scaling. For text samples, following the augmentation method of Gao et al. (2021) we use dropout masks.

- Neural architecture: For CIFAR-10 and CIFAR-100 we use ResNet-18 (He et al., 2016); for STL-10, TinyImageNet and ImageNet we use ResNet-50 (He et al., 2016); for the Wikipedia dataset we use BERT$_{\text{base}}$ (Devlin et al., 2019).

- Batch size and embedding dimension: for experiments in CIFAR-10 and CIFAR-100 we choose batch size 512, and for STL-10 we choose batch size 64 to accommodate one GPU training. Finally, for TinyImageNet and ImageNet, we choose batch size 256. For all the vision datasets, we choose the embedding dimension to be 512. Regarding the language dataset, the batch size is 64 with the feature dimension 768. In all of these cases, our assumption $N \le d + 1$ in Theorem 5 is satisfied.

- Hyperparameters: in all our experiments we fix the constant factor $\mu = 1$. We find that in practice the weight parameter $\alpha$ often needs to be large (*e.g.*, in the Wikipedia dataset), which requires moderate tuning. Note that we also implement RPC (Tsai et al., 2021) in our paper. For all the datasets, we follow Tsai et al. (2021) and choose the relative parameters $\alpha = 1.0, \beta = 0.005$ and $\gamma = 1.0$ for all datasets.

- Optimizer and learning rate scheduler: For vision tasks, we use SGD with momentum for optimization and the cosine learning rate scheduler (Loshchilov & Hutter, 2017). For the natural language task, we use Adam with weight decay (Loshchilov & Hutter, 2018) and the linear decay scheduler.

- Evaluation metric: for vision tasks, we use $k$-nearest-neighbor (kNN) and linear evaluation to evaluate the performance, based on the learned embeddings. For the NLP task, we use the Spearman's correlation to evaluate the averaged semantic textual similarity score (Gao et al., 2021).

- Baseline methods: for the four baseline methods, we follow the implementations in:
  - MoCo: `https://github.com/facebookresearch/moco`
  - SimCLR: `https://github.com/sthalles/SimCLR`
  - Uniformity: `https://github.com/SsnL/align_uniform`
  - RPC: `https://github.com/martinmamql/relative_predictive_coding`.

  For fair comparison we use the experimental settings in Table 5 for all the baseline methods, which might differ from the original settings.

Table 5: Detailed experimental settings. `arch`: the neural network architecture used. $N$: batch size; $d$: the dimension of the feature representation; `lr`: learning rate; $\mu$: the constant factor in $\mu$; $1/(2\sigma^2)$ and $\alpha$ follow from Algorithm 1; `epoch`: the number of epochs we run; $k$: the number of nearest neighbors in kNN evaluation.

| Dataset | `arch` | $N$ | $d$ | `lr` | $\mu$ | $(2\sigma^2)^{-1}$ | $\alpha$ | `epoch` | $k$ |
|---|---|---|---|---|---|---|---|---|---|
| CIFAR-10 | ResNet-18 | 512 | 512 | 0.1 | 1 | 1 | 40 | 800 | 200 |
| CIFAR-100 | ResNet-18 | 512 | 512 | 0.1 | 1 | 1 | 40 | 1000 | 200 |
| STL-10 | ResNet-50 | 64 | 512 | 0.1 | 1 | 1 | 40 | 800 | 200 |
| TinyImageNet | ResNet-50 | 256 | 512 | 0.1 | 1 | 1 | 40 | 800 | 200 |
| ImageNet | ResNet-50 | 256 | 512 | 0.1 | 1 | 1 | 40 | 100 | n/a |
| Wikipedia | BERT$_{\text{base}}$ | 64 | 768 | 3e-5 | 1 | 20 | 409600 | 1 | n/a |

Table 5 gives common choices of hyperparameters for different datasets. Note that we may need to further finetune $\alpha$ and $\sigma$ for different $f$-divergences. See our supplementary code for more details.

Table 6: Ablation study on weighting parameter $\alpha$ for KL and JS divergences on CIFAR-10. We compare test classification accuracies (%) for different choices of $\alpha$ using kNN evaluation.

| $\alpha$ | 0.1 | 1 | 10 | 20 | 30 | 40 | 50 |
|---|---|---|---|---|---|---|---|
| KL | 13.16 | 77.60 | 83.53 | 83.77 | 81.39 | **84.19** | 82.77 |
| JS | 8.84 | 73.31 | 81.39 | 83.21 | 83.49 | **84.06** | 82.61 |

Table 7: Ablation study on weighting parameter $\alpha$ for KL and Pearson $\chi^2$ divergences on Wikipedia. We compare the semantic textual similarity (STS) via the Spearman's correlation for different choices of weighting parameter $\alpha$.

| $\alpha$ | 1 | 10 | $10^2$ | $10^3$ | $10^4$ | $10^5$ | 409600 | $10^6$ |
|---|---|---|---|---|---|---|---|---|
| KL | 67.52 | 70.47 | 72.43 | 75.12 | 76.90 | 77.78 | **78.02** | 77.78 |
| Pearson | 64.58 | 67.78 | 71.58 | 74.03 | 74.95 | 74.40 | **77.59** | 76.47 |

## C.2 ADDITIONAL ABLATION STUDY ON WEIGHTING PARAMETER

We provide additional ablation study on the weighting parameter $\alpha$. We perform experiments using a vision dataset (CIFAR-10) and a language dataset (Wikipedia). For CIFAR-10, we vary $\alpha$ from 0.1 to 50 for KL and JS divergences and run for 200 epochs. Table 6 justifies our choice of $\alpha$ in Table 5, where the downstream test classification accuracy indicates the optimal performance when choosing $\alpha = 40$. For the Wikipedia dataset, we observe that a much bigger $\alpha$ is desirable for maximum performance. We vary $\alpha$ from 1 to $10^6$ for KL and Pearson $\chi^2$ divergences and run for 1 epoch, as there is a large number of samples ($10^6$) in the language dataset. Table 7 justifies our choice of $\alpha$ in Table 5, where the best performance is reached at $\alpha = 409600$. Such an $\alpha$ is found by starting from $\alpha = 100$ and doubling iteratively.

## C.3 ADDITIONAL EXPERIMENTS

Our final experiments show that $f$-MICL is stable in terms of training and the variation of performance is well controlled.

**Training stability** We depict the training loss curves of different divergences on CIFAR-10 in Figure 4. This figure shows that our methods exhibit stable training dynamics with fast convergence.

**kNN evaluation and additional $f$-divergences** We show more detailed results of Table 2 in Table 8, including experiments using $k$-nearest neighbour (kNN) evaluation. Additionally, we have added experiments on other $f$-divergences such as Squared Hellinger and Tsallis-$\alpha$ divergences.

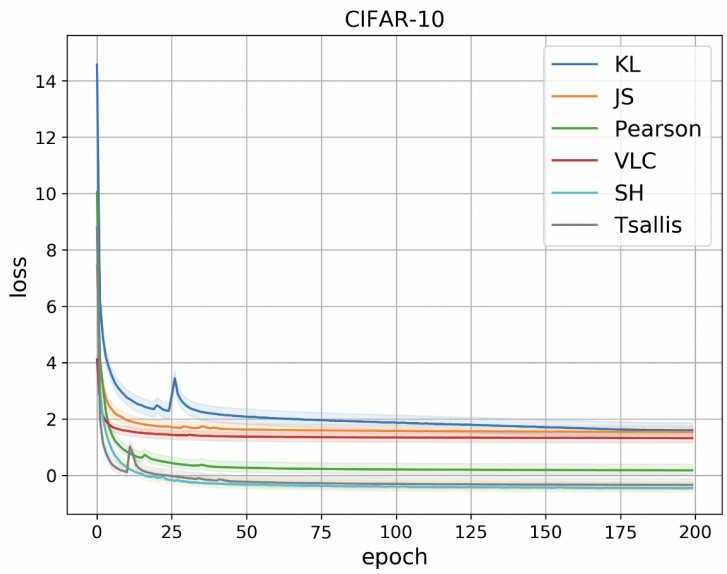

Figure 4: The training loss curves of various $f$-divergences on CIFAR-10 with 200 epochs.

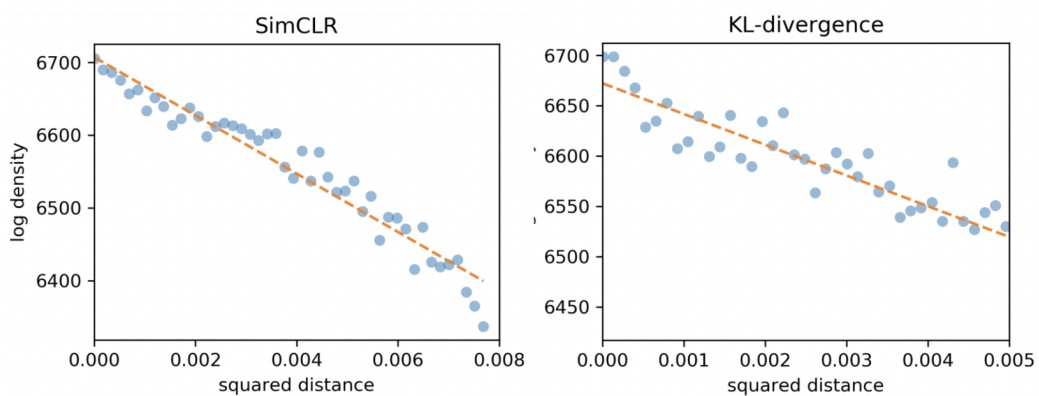

Figure 5: Experiment for verifying Assumption 3. We draw the relation between the squared distances $\|x^g - y^g\|^2$ and the averaged $\log p_g$ with RealNVP. The features are learned by different algorithms trained on CIFAR-10. (**left**) SimCLR; (**right**) $f$-MICL with the KL divergence.

Table 8: Test classification accuracy (%) on the vision datasets. For the Wikipedia dataset we evaluate the semantic textual similarity (STS) via the Spearman's correlation. For each method, we take three separate runs, and show the mean and stand derivation.

| Evaluation | Dataset | Baselines | | | | $f$-MICL | | | | | |
|---|---|---|---|---|---|---|---|---|---|---|---|
| | | MoCo | SimCLR | Uniformity | RPC | KL | JS | Pearson | SH | Tsallis | VLC |
| Linear | CIFAR-10 | 90.30 ±0.19 | 89.71 ±0.37 | 90.41 ±0.26 | 90.39 ±0.25 | **90.61** **±0.47** | 89.66 ±0.28 | 89.35 ±0.52 | 89.52 ±0.25 | 89.15 ±0.42 | 89.13 ±0.33 |
| | CIFAR-100 | 62.77 ±0.17 | 62.75 ±0.45 | 62.51 ±0.36 | 62.66 ±0.39 | 63.00 ±0.44 | **63.11** **±0.33** | 61.69 ±0.57 | 61.47 ±0.23 | 60.55 ±0.44 | 61.19 ±0.29 |
| | STL-10 | 83.69 ±0.22 | 82.97 ±0.32 | 84.44 ±0.19 | 82.41 ±0.14 | 85.33 ±0.39 | **85.94** **±0.17** | 82.64 ±0.37 | 82.80 ±0.27 | 84.79 ±0.34 | 83.27 ±0.72 |
| | TinyImageNet | 35.72 ±0.17 | 30.56 ±0.28 | 41.20 ±0.19 | 34.95 ±0.25 | 39.46 ±0.20 | **42.98** **±0.18** | 38.45 ±0.54 | 40.83 ±0.67 | 32.99 ±0.49 | 38.65 ±0.45 |
| | ImageNet | 58.59 | 57.66 | 59.12 | 56.11 | 58.91 | **61.11** | 55.33 | 52.37 | 53.11 | 54.26 |
| kNN | CIFAR-10 | 88.70 ±0.22 | 84.92 ±0.39 | 89.42 ±0.18 | 84.21 ±0.24 | 89.34 ±0.57 | 89.12 ±0.38 | **89.44** **±0.60** | 88.13 ±0.18 | 89.18 ±0.62 | 89.15 ±0.23 |
| | CIFAR-100 | 59.21 ±0.15 | 53.47 ±0.47 | 61.07 ±0.38 | 50.02 ±0.34 | 59.12 ±0.37 | **61.36** **±0.35** | 58.68 ±0.57 | 57.66 ±0.33 | 58.12 ±0.45 | 59.17 ±0.28 |
| | STL-10 | 78.77 ±0.25 | 74.34 ±0.14 | 79.57 ±0.52 | 73.27 ±0.40 | 79.99 ±0.47 | **80.45** **±0.19** | 76.64 ±0.26 | 78.31 ±0.33 | 76.11 ±0.24 | 79.34 ±0.62 |
| | TinyImageNet | 36.22 ±0.20 | 29.60 ±0.39 | 37.44 ±0.27 | 24.25 ±0.35 | 36.17 ±0.29 | **38.20** **±0.26** | 35.14 ±0.63 | 35.56 ±0.77 | 33.11 ±0.52 | 35.21 ±0.33 |
| STS | Wikipedia | 77.88 ±0.15 | 77.40 ±0.12 | 77.95 ±0.08 | 68.32 ±0.23 | **78.02** **±0.13** | 76.76 ±0.09 | 77.59 ±0.12 | 73.60 ±0.10 | 72.68 ±0.09 | 55.07 ±0.13 |

**Verification of Assumption 3** Throughout our paper we made an assumption (Assumption 3) that the joint feature distribution is a Gaussian kernel. However, is it a valid assumption? In this experiment we try to show some empirical evidence that this assumption approximately holds in practice. Recall that Assumption 3 says that the joint feature distribution of positive pairs is:

$$p_g(x^g, y^g) \propto \exp\left(-\frac{\|x^g - y^g\|^2}{2\sigma^2}\right) \tag{C.1}$$

if the RBF kernel is Gaussian. In order to estimate the joint density of positive pairs, we use normalizing flows (Dinh et al., 2017), which is a popular method for the density estimation. Popular normalizing flow models include RealNVP (Dinh et al., 2017), NICE (Dinh et al., 2014) and Glow (Kingma & Dhariwal, 2018). Equation C.1 is equivalent to the following:

$$\log p_g(x^g, y^g) = -\frac{\|x^g - y^g\|^2}{2\sigma^2} + \text{const}, \tag{C.2}$$

and thus it suffices to show that the log likelihood is linear w.r.t. the distances between each positive pair. In Figure 5, we plot the relation between $\log p_g$, estimated by RealNVP[1], and the squared distances $\|x^g - y^g\|^2$. The representations are learned by SimCLR, and $f$-MICL with the KL divergence on the CIFAR-10 dataset. To alleviate the estimation error in the flow model, we divide the distances into small intervals and compute the average log likelihood within each interval. From this figure we can see that the log likelihood is roughly linear w.r.t. the squared distance, and thus verifying our Assumption 3.

---

[1] Code available at https://github.com/ikostrikov/pytorch-flows.

