# OpenReview forum: "$f$-Mutual Information Contrastive Learning"
_ICLR.cc/2022/Conference — ICLR 2022 Submitted_

### Official Review · Reviewer_HSE4 · 2021-11-02

**Correctness:** 3
**Technical Novelty And Significance:** 3
**Empirical Novelty And Significance:** 3
**Recommendation:** 6
**Confidence:** 3

**Main Review:**

It is novel to substitute the cosine similarity with $f$-divergence in contrastive learning, and the experiments validate that this is a very effective modification.

Theorem 5 is an important condition to help us choose the correct function which can avoid feature collapse.

The notation in Figure 1 is not easy to understand. I suggest the authors think of another way to express $-f^*f'$.

The description of the algorithm is not well written, according to Figure1, there is aug1, aug2, and aug. I assume aug is equal to either aug1 or aug2. And how to randomly sample augmentation functions t1 and t2?

**Summary Of The Paper:**

This paper proposed to combine contrastive learning with $f$-divergence, which naturally attains the alignment for positive pairs and the uniformity for data representations. The authors provided some theoretical results on choosing the correct function f and the upper bound of the estimation error. Finally, the authors carried out extensive experimental results to validate the effectiveness of the method.

**Summary Of The Review:**

Theory 5 provides a way to tell whether an f divergence could be a good candidate, which attracts me a lot. But at the same time, the description of the main algorithm is not complete, some details are missing and the notations in the figure is not well-explained.

---

> ### Author Response · Authors · 2021-11-22
> **Response to Reviewer HSE4**
>
> Thank you for your positive feedback and your suggestions on Figure 1. We have modified this Figure and added explanations in the caption (colored as blue).
>
> > **Q1: Regarding the notations in Figure 1.**
>
> We added the explanation of the notations in the caption of Figure 1. $f^*$ denotes the Fenchel conjugate and $f'$ denotes the derivative.
>
> > **Q2: Regarding the notations ${\tt aug}, {\tt aug}_1, {\tt aug}_2$: are they the same augmentations? How to randomly sample augmentation functions $t_1$ and $t_2$?**
>
> Thanks for the comment. We have modified Figure 1 accordingly. These augmentations are drawn from the same family of augmentation, which is a composition of random resized crop, random horizontal flip, color jittering and gray scaling, following the same setting as SimCLR (see our Appendix C.1).

---

### Official Review · Reviewer_fZBZ · 2021-11-02

**Correctness:** 4
**Technical Novelty And Significance:** 3
**Empirical Novelty And Significance:** 3
**Recommendation:** 6
**Confidence:** 4

**Main Review:**

Pros:
1) The paper is well written and easy to understand.
2) The experiment is quite comprehensive, and the results look positive.

Cons:
1) The paper has limited novelty compared to the InfoNCE framework.
2) The advantages/disadvantages of the proposed method compared to InfoNCE are not well analyzed in the paper.
3) The standard deviation of results is not provided, results of the baseline do not match those in the original paper.

**Summary Of The Paper:**

The paper proposes another variant of Contrastive Learning that uses a different lower bound of the mutual information based on f-divergence instead of InfoNCE.

**Summary Of The Review:**

1) Novelty:
- The main limitation of the paper is that the proposed idea is not novel. The variational lower bound of mutual information (MI) based on f-divergence was well discussed in previous works (e.g., [1, 2, 3]) so it is straightforward if we replace the InfoNCE bound with the new bound.

2) About the method:
- The main advantage of InfoNCE compared to other lower bounds of MI is its low variance which allows stable learning to achieve good representations. The low variance comes from the fact that InfoNCE uses multiple negative samples per positive sample and normalizes over them. On the downside, InfoNCE is biased and requires a large batch size. SimCLR has this drawback but it was overcome by MoCo. Therefore, I expect the authors to theoretically and empirically analyze the advantages and disadvantages of their proposed method compared to InfoNCE in terms of bias-variance trade-off. They should also compare their method with MoCo, which has been shown to perform better than SimCLR.

3) About experiments:
- The experiment results do not show the clear advantage of the new method compared to the existing baselines based on InfoNCE when using a linear classifier. The difference in performance in Table 3 is small which can be attributed to randomness in running. I think the authors should at least provide the standard deviation of their results to highlight the significance.

- The results in Table 3 in the paper are still very far from the results reported by SimCLR (Table 8 in the SimCLR paper). For example, SimCLR reports 95.3% accuracy on CIFAR10 with a linear classifier but in Table 3, the authors only show 89.7% accuracy. I think the authors should check their experimental settings again to make their results more comparable.

[1] On Variational Bounds of Mutual Information, Poole et al., ICML 2019

[2] Mutual Information Neural Estimation, Beghazi et al., ICML 2018

[3] Learning Deep Representations by Mutual Information Estimation and Maximization, Hjelm et al., ICLR 2019

---

> ### Author Response · Authors · 2021-11-22
> **Response to Reviewer fZBZ**
>
> Thank you for your comments and your suggestions to improve our paper. Please see our point-to-point response below:
>
> > **Q1: The paper has limited novelty compared to InfoNCE. $f$-divergence with MI has been discussed in previous works (e.g. [1, 2, 3]).**
>
> Thanks for pointing out these references. We took a detailed look at [1, 2, 3] and found that our $f$-MICL framework is indeed novel:
> - The lower bounds in reference [1] are Donsker--Varadhan, InfoNCE, and NWJ. None of these include the variational lower bound as our eq. (2), although the NWJ is a special case. Besides, their experiments are representation learning of 2d distributions, which is far from our contrastive learning of large-scale image datasets.
> - Reference [2] uses Donsker--Varadhan lower bound for KL, which is different from our variational lower bound for general $f$-divergences. Besides, their task is improving GAN training with MI, which is far from our contrastive learning of large-scale image datasets.
> - Reference [3] is mostly close to contrastive learning. However, we have explicitly discussed the differences in Section 6. Reference [3] uses the concatenation critic as the similarity, whereas we use the Gaussian similarity. Our similarity function naturally leads to *alignment* and *uniformity*.
>
> We would be happy to discuss these references in a revised version if the reviewer feels necessary.
>
> > **Q2: I expect the authors compare with InfoNCE in terms of bias-variance tradeoff.**
>
> Thank you for your suggestion, and we are happy to make the comparison regarding the bias-variance tradeoff following your comments. Similar to SimCLR, for each sample, we use multiple negative samples per positive sample, and thus the variance is as small as SimCLR. Meanwhile, we found that our bias term is smaller than that in SimCLR. This is verified by our experiments in Table 2. In addition, on the right panel of Figure 3, we show that we can achieve the same performance with a smaller batch size than SimCLR.
>
> > **Q3: The authors should compare their method with MoCo.**
>
> Thanks for the suggestion and we have added experiments for MoCo for all datasets in Table 2. This comparison shows that our algorithm still has a clear advantage over MoCo.
>
> > **Q4: The experiments do not show a clear advantage over baselines. The authors should at least provide standard deviation to highlight the significance.**
>
> We have added the standard deviations for every optimal divergence for all our datasets (except ImageNet due to the limited time) to show the real advantage of our objectives in Table 2 and Table 8. From this table we can see that for the linear evaluation, our method is advantageous with statistical rigor, especially for large datasets like ImageNet (nearly 2\% gain).
>
> > **Q5: The results are still far from the results reported by SimCLR.**
>
>  Thank you for mentioning that, we would like to clarify two differences in terms of the experimental set-up:
> - For the accuracy results on CIFAR-10, Table 8 in SimCLR considers the task of transfer learning, where the pretrained models are obtained by training on *ImageNet*, while our task uses pretrained models obtained directly from *CIFAR-10*, which is much smaller.
> - Table 8 in SimCLR applies ResNet-50 as the encoder architecture, whereas we use a smaller architecture ResNet-18.
>
> Due to these two differences, it is not surprising that our reported results are worse than the ones reported in [4]. For fair comparison we run SimCLR, as well as other baselines, using the same experimental settings as in Table 5.
>
> [4] Ting Chen, Simon Kornblith, Mohammad Norouzi, and Geoffrey Hinton. A simple framework for contrastive learning of visual representations. In ICML 2020.

---

> > ### Comment · Reviewer_fZBZ · 2021-11-25
> > **Adjust my score based on the authors' response**
> >
> > Thank the authors for their response. The authors have addressed my concerns so I raise the score to 6.

---

### Official Review · Reviewer_Q5hx · 2021-11-02

**Correctness:** 4
**Technical Novelty And Significance:** 3
**Empirical Novelty And Significance:** 3
**Recommendation:** 6
**Confidence:** 4

**Main Review:**

Strength:

The theoretical results show that the generalization from mutual information to f-mutual information is not merely a mathematical game but also has its own merits. Specifically:

1. The natural connection with alignment and uniformity is a neat observation.

2. The finite-sample result on uniformity provides a counterpart of the existing results in the infinite-sample case, which is interesting because it demonstrates that not all f-divergences are equal.

3. The result of the estimation error is able to handle the dependent negative pairs, using a non-trivial application of Rademacher complexity theory.

Weakness:

1. The results in Table 9 in the Appendix seem not to demonstrate a clear advantage of the new framework compared with existing baselines. As such, I am worried if the results in the main body, such as  Table 2, are convincing enough to illustrate the advantage of the new framework.

2. For f-divergences that satisfy the condition (12) in Theorem 5, can you provide some guidelines on how to choose between them?




**Summary Of The Paper:**

This paper proposes the f-mutual information objective for contrastive learning, which generalizes the existing mutual information framework. Theoretically, it makes a connection with alignment and uniformity in Wang & Isola (2020) when the joint feature distribution can be represented by a radial basis function, and derives its estimation error. Numerically, it demonstrates the efficacy of f-mutual information with Gaussian similarity.

**Summary Of The Review:**

The paper is well-motivated and clearly written. I enjoy reading the paper, and I believe the results are technically sound as I do not find major flaws in the proof. Although mathematically the new framework simply replaces mutual information with f-mutual information, I think it produces several interesting observations and draws connections with the literature. Due to computational efficiency, the authors are not able to provide confidence intervals of the objective in their numerical experiments, which raises the concerns whether the new framework is indeed better in some applications.

---

> ### Author Response · Authors · 2021-11-22
> **Response to Reviewer Q5hx**
>
> Thank you so much for your positive feedback and your summary of our contributions. We address your questions/concerns below:
>
> > **Q1: In Table 9, $f$-MICL does not seem to demonstrate a clear advantage over baselines.**
>
> To make sure that our improvement is not simply statistical fluctuation, we have added error bars for every divergence for all our datasets (except ImageNet due to the limited time) to show the real advantage of our objectives in Table 2 and Table 8. From these tables, we can see that although $f$-MICL does not have a large margin over baselines for CIFAR-10, for large-scale experiments such as ImageNet, our method has a noticeable margin over baselines (nearly 2%). Moreover, this improvement is more significant with linear evaluation, which is more practical for larger datasets.
>
> > **Q2: Can you provide some guidelines for how to choose $f$-divergences in Table 1?**
>
> Based on our experiments in Table 2, JS-divergence performs better in most cases than other $f$-divergences satisfying Theorem 5. In practice, to pick a proper $f$-divergence for a particular dataset, we can use a separate validation set to choose among these divergences.
>
> > **Q3: The authors are not able to provide confidence intervals of the objective.**
>
> We have added standard deviations for every divergence for all our datasets (except ImageNet which is quite expensive to run in a limited time) to show the real advantage of our objectives in Table 2 and Table 8.

---

### Official Review · Reviewer_qEh5 · 2021-11-04

**Correctness:** 2
**Technical Novelty And Significance:** 2
**Empirical Novelty And Significance:** 3
**Recommendation:** 5
**Confidence:** 4

**Main Review:**

This paper addresses an interesting problem, and demonstrates useful empirical results. While the paper has promise, I cannot recommend acceptance of the current state of the paper due to sometimes incoherent results, unclear statement of contributions, and some unjustified assumptions. Of course, I am happy to change my score if I mis-understood any parts of the paper or if the issues are improved in the rebuttal period.

Theoretical: I find the assumptions quite odd. If the paper already assumes that the true distribution over features is a Gaussian, then naturally Gaussian kernels are the right choice. Therefore, I do not really see why the theorem provides interesting insight. Such a strong assumption might require experimental / theoretical justification, which is not provided in the paper.

In section 4 the paper bounds the estimation error of the mutual information. However, I do not see how this is relevant to the goal of representation learning. If accurate estimation of the lower bound matters for representation learning, I think the paper needs to demonstrate this either theoretically or empirically.

These are some claims in the intro that in my opinion are not adequately supported in the paper, and hence might benefit from being downplayed. One seeming contradiction is that the intro first claims that a tighter bound on the MI does not improve the representations, which is one of the motivations for this work. However, much of the theoretical results focus on how to obtain tighter bounds on the f-MI. This can be confusing as the paper did not justify why tighter bound to Shannon MI leads to worse representation learning performance, but it is desirable to obtain tighter bounds to the f-MI (which after all includes Shannon MI as a special case)? Also, the introduction claims that the newly proposed similarity function is ‘optimal for practical use’. However, I do not think such claims are adequately supported by theoretical or empirical results.

Empirical: Given that the theoretical results are not fully convincing, the experiments become crucial. The paper would be a great contribution if it can convincingly demonstrate the empirical advantage of Gaussian similarity compared to cosine similarity, and the benefit of using another f-MI instead of Shannon MI. The experiments provide some evidence that some choices of f-MI (JS) is better on CIFAR, STL and TinyImagnet compared to baselines such as SimCLR. In addition, the experiments provide some evidence that using Gaussian similarity is better than using cosine similarity.

The experiments have some short-comings. First, there is no experiment on the original setup of e.g. SimCLR which was applied to Imagenet. Second, on the STS dataset, compared to the cited paper, the reported baseline performance seems worse? While this is not a fatal issue, maybe the original paper used additional techniques that were difficult to reproduce, it is certainly something that needs to be discussed. Finally, there is not insufficient statistical rigor such as error bars, as some of the improvements are small enough to be possible to come from random fluctuations.

That being said, I do think that the paper can become much stronger with a clearer statement of the actual contribution. In my opinion, the most interesting contribution would be proposing that f-MI lead to a new class of new objectives, that while not necessarily better than Shannon-MI, at least provides more options. In addition, experiments can be used to study which options are the best empirically.

--------------

Post rebuttal: Thank you for the detailed rebuttal. After the rebuttal, I feel better about the experiments. There is a lot of subtlety with representation learning evaluation, so I believe the paper should include all the details of evaluation, including justification on why e.g. SBERT-base is chosen as a baseline (which is fine as the baseline does not necessarily have to be state-of-the-art with all the bells and whistles, but it should be clearly discussed and disclosed since the original paper contains methods with significantly better performance). The same goes for the newly added Imagenet experiments.

In addition, while the authors understandably had insufficient time to revise the paper during the rebuttal period to address the writing issues, I believe the claims of the paper should be more precise and well-supported by the actual experiments. A mediocre contribution that is well supported is better science and probably contributes to research more than amazing contributions that are weakly supported or questionable.

Given the detailed rebuttal and the new experiments, I will raise my score.

**Summary Of The Paper:**

This paper makes two contributions to contrastive representation learning: 1. Using the more general f-mutual information rather than using Shannon mutual information for contrastive learning 2. Experimental results to compare the possible options given the new design freedom.

**Summary Of The Review:**

The paper proposes an interesting family of new contrastive learning objectives, and some empirical comparison of the objectives. However, I think the paper needs a more clear and accurate statement of the contribution, better justified theoretical results, and better empirical evaluations.

---

> ### Author Response · Authors · 2021-11-22
> **Response to Reviewer qEh5 (Q1-Q2)**
>
> Thank you for your detailed comments and various suggestions for us to improve the current draft. We would like to clarify your concerns below:
>
> > **Q1: It is an odd assumption that the joint feature distribution is Gaussian. What is the insight of the theorem? Can you justify the assumption experimentally/theoretically?**
>
> Thank you for the great question. We admit that with the joint feature distribution as a Gaussian, the derivation from Lemma 2 to Theorem 4 is natural. The importance of Theorem 4 lies in the fact that it offers more choices of similarity functions, rather than the *de facto* cosine similarity in contrastive learning. If we *knew* what the joint feature distribution should look like in the end, we would be able to design better similarity functions.
>
> We understand your concern that this assumption might be strong. To address this concern, we have conducted additional experiments to verify it in Appendix C.3 (see Figure 5 and the following detailed analysis). Briefly speaking, to verify Assumption 3 we aim to show that the log joint density is linear with the squared distance of positive features (see Equation C.2).
>
> To estimate the joint density we adopted RealNVP, which is a popular flow-based model. In Figure 5, we used the obtained positive features on CIFAR-10 for the verification. With the $x$-axis representing the squared distance and the $y$-axis representing the average log-density, we can observe an approximately linear curve, which empirically demonstrates our assumption. As a comparison, we also showed the squared distance v.s. the average log-density for SimCLR in Figure 5, and observed the same linear relationship. This may indicate the generality of Assumption 3 in other existing contrastive learning work.
>
> > **Q2: How is the estimation error relevant to the goal of representation learning?**
>
> Our Theorem 6 gives the *sample complexity* for contrastive learning, i.e., how many samples we need to have a small estimation error. This is directly relevant to representation learning since we always use a finite number of samples to estimate the true distribution. If the number of samples, or the batch size is small, then the objective with finite sample estimation (eq. (10)) would not work due to the large estimation error. Our Theorem 6 gives theoretical justification for it. Moreover, in our experiments (Figure 3 right panel) we also find the drop of performance with smaller batch size.
>
> In fact, sample complexity has been a central topic in machine learning theory [1] for a long time. In self-supervised learning, however, such a study is scarce due to the non-i.i.d. property of sample pairs. Theoretical results for sample complexity are strongly necessary (see https://sslneurips21.github.io/). Our result, as also acknowledged by **Reviewer Q5hx**, is non-trivial due to the dependent negative pairs, which is common in contrastive learning [2, 3].
>
> Last but not least, our Theorem 6 gives the tradeoff between approximation and estimation. If the function class is very complicated, e.g., a very large neural network architecture like a vision transformer, then the estimation error is large but the approximation error is small. In contrast, if the function class is simple, e.g., a small neural network like three-layer MLP, then the estimation error is small but the approximation error is large. This gives us guidance that we should choose the function class with moderate complexity.
>
> [1] Shai Shalev-Shwartz and Shai Ben-David, Understanding machine learning: From theory to algorithms. Cambridge university press, 2014.
>
> [2] Ting Chen, Simon Kornblith, Mohammad Norouzi, and Geoffrey Hinton. A simple framework for contrastive learning of visual representations. In ICML 2020.
>
> [3] Kaiming He, Haoqi Fan, Yuxin Wu, Saining Xie, and Ross Girshick. Momentum contrast for unsupervised visual representation learning. In CVPR, 2020.

---

> ### Author Response · Authors · 2021-11-22
> **Response to Reviewer qEh5 (Q3-Q7)**
>
> > **Q3: The intro claims that a tighter bound does not improve the representations, but we try to find a tighter bound.**
>
> Thank you for pointing out this paradox and sorry for the confusion. We took a detailed look at [4] and [5] and here is the clarification:
> - Both papers give empirical evidence showing that maximizing MI may not always be ideal, but there are no theoretical results. In contrast, we can *theoretically* justify that maximizing the $f$-MI lower bound gives alignment and uniformity (Section 3.2). Their settings are also different from ours and thus their conclusions may not directly apply. For example, in [4], the similarity function is bilinear critic $f(x, y) = x^\top W y$, while we use the Gaussian similarity. In [5] they considered NCE loss which is different from our $f$-MICL objective.
> - As pointed out in Sec 3.1, [4] , maximizing MI may either improve or worsen the performance, depending on the architecture. Our Theorem 5 is along this line: maximizing the MI estimator might not always help, depending on the $f$-divergence we choose.
> - [5] shows that maximizing the MI between two views can learn task-irrelevant information and thus worsen the performance. To overcome this difficulty, SimCLR [2] uses a stronger class of augmentation so that the task-irrelevant information cannot overfit. Our work follows along the line of SimCLR.
>
> We will revise the draft carefully after the rebuttal to adopt the clarification if the reviewer finds it necessary.
>
> [4] Michael Tschannen, Josip Djolonga, Paul K. Rubenstein, Sylvain Gelly, and Mario Lucic. On mutual information maximization for representation learning. In ICLR, 2020.
>
> [5] Yonglong Tian, Chen Sun, Ben Poole, Dilip Krishnan, Cordelia Schmid, and Phillip Isola. What makes for good views for contrastive learning. In NeurIPS 2020.
>
> > **Q4: Why Gaussian similarity is optimal for practical use?**
>
> Sorry for the confusion. What we meant is that the Gaussian similarity is optimal if the joint feature distribution is a Gaussian kernel. Also, as we have shown in Appendix C.3, this assumption approximately holds on image datasets such as CIFAR-10.
>
> > **Q5: There is no experiment on the original setup e.g. SimCLR applied to ImageNet.**
>
> We have added the original setup of the baselines, as well as our $f$-MICL results on ImageNet. Please see the updated Table 2 for details.
>
> > **Q6: On the STS dataset, the reported baseline seems worse.**
>
> Compared to the cited paper, our reported STS baseline is actually *better*. We apply the same SimCSE-BERT ${\tt base}$ model [6] for our STS experiment (see the implementation details in Section C.1 in Appendix C), and report the averaged STS score 77.40 (SimCLR baseline) in our Table 2. This is indeed better than the 76.25 average STS score in Table 5 (Unsupervised models, SimCSE-BERT ${\tt base}$) in SimCSE [6].
>
> [6] Tianyu Gao, Xingcheng Yao, and Danqi Chen. SimCSE: Simple contrastive learning of sentence embeddings. In EMNLP 2021.
>
> > **Q7: There is not sufficient statistical rigor such as error bars.**
>
> We have added error bars for every divergence for all our datasets (except ImageNet due to the limited time) to show the real advantage of our objectives in Table 2 and Table 8.
>
> Thank you again for your valuable suggestions and your encouragement. We hope our revision could alleviate some of your concerns.

---

### Author Response · Authors · 2021-11-22
**General response**

We thank all reviewers for their careful reading of our paper and their suggestions. To address the concerns from reviewers, we have made the following main changes in our draft, colored as blue:
- We added the standard deviations of all our previous experiments in Table 2, and the full version can be seen in Table 8. This shows that our $f$-MICL is better than popular baselines with statistical rigor.
- We added the experiments on MoCo in Table 2. Even though MoCo performs better than SimCLR, our $f$-MICL still shows a clear advantage over MoCo, especially on large datasets like ImageNet (more than 2\%).
- We added experiments on ImageNet in Table 2, which shows that our $f$-MICL is better than SimCLR by nearly 3.5\%. This verifies the practicability of $f$-MICL on large datasets.
- We empirically verified Assumption 3 by estimating the joint feature distribution using a flow-based model (RealNVP).
- We modified Figure 1 and changed the caption to make it clearer.
- We added the reproducibility statement on Page 10.

---

### Author Response · Authors · 2021-11-29
**Any questions before the end of the discussion period?**

We would like to thank again all reviewers.

Please let us know if there are additional questions or concerns before the end of the discussion period. We would be happy to discuss or address any additional comments.

---

### Decision · Program_Chairs · 2022-01-20

**Decision:**

Reject

**Comment:**

Most of the reviewers have concerns that the experimental results don’t show stronger enough improvements over baselines and that the theoretical contribution of the paper is not completely clear. These concerns make the paper a borderline paper for NeurIPS. Some reviewers have pointed out problematic or unsupported claims in the paper. With these in mind, I encourage the authors to revise the paper with more clarity and address the reviewers' comments on the exposition of the paper.